# Self-Supervised Dynamical System Representations for Physiological Time-Series

Yenho Chen [1]  Max Xu [2][3]  James M. Rehg [2]  Christopher J. Rozell [4]

## Abstract

Self-supervised learning for physiological time-series aims to captures the identity of the underlying dynamical process while filtering irrelevant noise. However, existing approaches may obscure the clinical semantics important for downstream transferability. Weakly constrained pretext tasks (i.e. contrastive learning, MAE) may incorrectly ignore the underlying dynamical structure, while structurally constrained models (i.e. SVAEs) are unable to selectively filter sample-specific noise. To bridge this gap, we propose **PULSE**, a novel pretraining objective that simultaneously preserves dynamical relationships important to physiological time-series while selectively removing irrelevant noise. We achieve this by formulating a dynamical systems model to identify transferable and non-transferable information between time-series windows, and target the former through a novel cross-reconstruction objective. We establish theory that provides conditions for when transferrable information is recovered, and empirically validate it through synthetic experiments. On several real-world datasets, PULSE effectively distinguishes clinical semantic classes, increases label efficiency, and improves transfer learning performance.

## 1. Introduction

Self-supervised learning (SSL) is a powerful framework for learning general-purpose representations from unlabeled physiological time series. These representation can be used to track physiological states, detect diseases, and improve our understanding of biology (Perochon et al.; Chen et al., 2023b; Li et al., 2025a).

To learn these representations, many time-series SSL strategies have emerged to selectively preserve information relevant to the identity of the underlying physiological process while filtering out irrelevant noise (Tian et al., 2020). These methods can be categorized into those with weakly constrained pretext tasks that prioritize downstream transferability, such as Contrastive Learning (CL) and Masked Autoencoding (MAE) (Xu et al., 2024; Geenjaar and Lu, 2025), and those with structurally constrained pretext tasks that recover specific components of a factorized latent generative process, such as Sequential Variational Autoencoders (SVAEs) (Sedler and Pandarinath, 2023).

These approaches have complimentary strengths and limitations. While weakly constrained approaches include explicit mechanisms to isolate signal from specific sources of noise, the lack of structural constraints may lead the model to inadvertently learn incorrect relationships or incorrectly discard clinically important signal components. For instance, CL explicitly removes noise information to obtain representations that are invariant across positive pairs (Chen et al., 2020). Yet, positive pairs formed through common augmentations (e.g. jittering, scaling) may change a signal's clinical identity which can incorrectly collapse samples with different clinical diagnoses. Similarly, while the MAE objective filters information that is not shared across masked views (Kong and Zhang, 2023), standard masking strategies allow the model to use future context to reconstruct past segments, which may prioritize non-causal spurious relationships over those that reflect the causal dynamics of physiological processes.

Conversely, structurally constrained models such as SVAEs learn a latent dynamical systems model that explicitly preserves causal temporal dependencies that are essential for describing individual physiological time-series windows (Girin et al., 2022). However, these models lack a mechanism for selectively removing sample-specific noise and instead may compress the signal arbitrarily (Ren et al., 2018), limiting their transferability across clinically re-

[1]ML@GT, Georgia Institute of Technology, Atlanta, Georgia [2]Health Care Engineering Systems Center, University of Illinois Urbana-Champaign, Champaign, Illinois [3]Google [4]School of Electrical and Computer Engineering, Georgia Institute of Technology, Atlanta, Georgia. Correspondence to: Yenho Chen <yenho@gatech.edu>, Christopher Rozell <crozell@gatech.edu>.

*Proceedings of the $43^{rd}$ International Conference on Machine Learning*, Seoul, South Korea. PMLR 306, 2026. Copyright 2026 by the author(s).

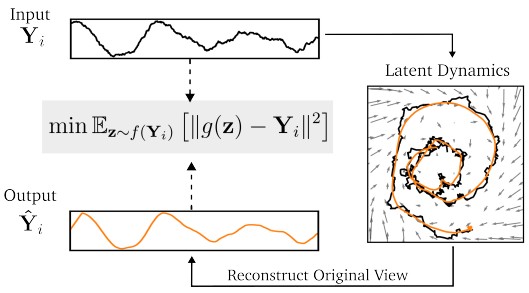

A) SVAE trains on direct reconstruction

B) PULSE trains on cross-reconstruction

*Figure 1.* Comparing Dynamical System Representations. **(A)** SVAE couple system estimation and reconstruction to the same input, which can confound the underlying dynamics with noise. **(B)** PULSE decouples these processes by estimating a system from $\mathbf{Y}_i$ that reconstructs augmented views $\mathbf{Y}_j \sim \mathcal{T}(\mathbf{Y}_i)$, forcing the model to capture shared dynamics and discard sample-specific noise.

lated time-series. This is due to the autoencoding objective (Alemi et al., 2018), which penalizes *any* deviation from the raw input. As a result, the model may encode noise (i.e. recording offsets, transient fluctuations), potentially obscuring clinically relevant patterns.

To bridge weakly and structurally constrained methods, we introduce a novel pretraining approach that simultaneously preserves temporal dependencies through a latent dynamical systems model while selectively removing sample-specific noise called **PULSE** (**P**hysiological self-s**U**pervised **L**earning using **S**ystem **E**ncoders)[1]. In doing so, we avoid the issues of weakly constrained pretext tasks while also improving the transferability of structurally constrained representations. Unlike SVAEs which model dynamics *within* individual time-series, PULSE models consider generative structure *between* similar time-series. By modeling mutliple similar time-series, we reveal sources of information that should be kept and discarded during pretraining. This description leads to our key insight: *system information* related to the generative parameters should be preserved since it is transferrable between independent time series produced by the same process, whereas *sample-specific information* about factors that are unique to each sample, such as initial conditions and process noise, is non-transferrable and should be discarded. As illustrated in Figure 1, we target this system information through a novel cross-reconstruction task that explicitly removes sample-specific noise by decoupling system estimation from reconstructed samples. In several synthetic and real datasets, we show how this leads to performance consistent improvements over existing baselines across several downstream tasks. Our contributions are summarized as follows:

1. This is the first work to use dynamical systems within a cross-reconstruction framework to model dependencies

between time-series samples and to selectively separate transferable information from noise.

2. We explore theory for PULSE that provides conditions for when system information is recovered and empirically validate this theory with a synthetic experiment.

3. In many real-world datasets, PULSE achieves SOTA performance, outperforming competitive baselines in linear evaluation, label efficiency, and transferability.

## 2. Background and Related Work

**Self-Supervised Learning for Time Series.** CL and MAE are the most studied weakly constrained paradigms for time-series due to their success in computer vision (Gui et al., 2024; Li et al., 2024; 2025b). In CL, the design of positive pairs determines what information the representation is invariant to (Tian et al., 2020). Positive pairs may be formed via augmentations (i.e. scaling, shifting, or jittering in SimCLR (Chen et al., 2020)), through sampling strategies (i.e. selecting time-neighbors in TNC (Tonekaboni et al., 2021), or overlapping crops in TS2vec (Yue et al., 2022)), or through a learned reconstruction measure as in REBAR (Xu et al., 2024). When applied to physiological time-series these methods may yield inconsistent performance across datasets (Liu et al., 2024) due to false positive pairs that incorrectly collapse different clinical states together. In MAE, the masking strategy filters out information not shared between complementary masked views (Kong and Zhang, 2023). For instance, block masking in TimeMAE (Cheng et al., 2026) or patch masking with channel independence in PatchTST (Nie et al., 2022) to capture shared information across masked windows. In contrast, channel masking (Wang et al., 2024) captures inter-channel dependencies. However, these masking strategies often treat time-series as images or language tokens, and ignore the temporal constraints of physiological systems. This allows for the possibility to exploit spurious shortcuts rather than learning representations related to the

---

[1]Code available at: https://github.com/yenhochen/PULSE

underlying dynamics. In PULSE, we avoid these limitations by imposing structural dynamical systems constraints that explicitly preserves important temporal relationships.

**Dynamical Systems Models of Physiological Signals.** Many physiological time-series arise from physical and biochemical processes governed by dynamical systems. Thus, these systems provide a shared modeling framework for characterizing the sources of information available in physiological signals. One prominent approach for modeling time-series generative processes are the *state-space models* (SSMs). In this framework, physiological time series are described as the evolution of a latent state $\mathbf{x}_{t_0} \in \mathbb{R}^n$ initialized at time $t_0$ that evolves forward through a dynamics function $\mathbf{x}_{t+1} = g_x(\mathbf{x}_t; \Theta)$, parameterized by system variables $\Theta$. This latent process can then produce measurements $\mathbf{y}_t \in \mathbb{R}^m$ through an observation function $\mathbf{y}_t = g_y(\mathbf{x}_t)$. Although dynamical systems models have been used to study a wide range of physiological processes, including cardiac dynamics (Bianco et al., 2025) and brain activity (Chen et al., 2024; Mudrik et al., 2024), these applications focus on within-sample structure and lack a mechanism to selectively filter sample-specific information within an SSL objective. In this work, we extend dynamical systems models to describe relationships between time-series samples, which leads to an explicit mechanism for selectively removing sample-specific noise during pretraining.

**Sequential Variational Autoencoders.** SVAEs leverage the SSM to learn representations corresponding to components of a dynamical systems. Training proceeds via an autoencoding task by maximizing the ELBO for an encoder-decoder network structure to enforce specific generative assumptions. For instance, LFADS (Sussillo et al., 2016) encodes each time series into an latent $\mathbf{x}_{t_0}$ under the assumption that all time-series evolve under a shared dynamics function. DSVAE (Yingzhen and Mandt, 2018) assumes that the dynamics is data-dependent and can be factorized into static and dynamic generative factors. Unlike CL and MAE, which have mechanisms for selectively filtering out noise, SVAEs methods may remove noise arbitrarily. Because the autoencoding task does not distinguish between transferrable signal from noise in the generative model, it instead encourages explaining all observed variability in the data. Consequently, the learned representations can be overly sensitive to irrelevant information, reducing their transferability to downstream tasks.

**Mixed-Effects Modeling.** One way to formulate graphical models for time-series data is the hierarchical, mixed-effects approach, where fixed effects capture shared dynamics and random effects capture individual-specific variability (Zhang et al., 2020; Xiong et al., 2019; Roques et al., 2025). However, these methods typically require the number of global and local components to be specified in advance, which can be difficult to determine in practice.

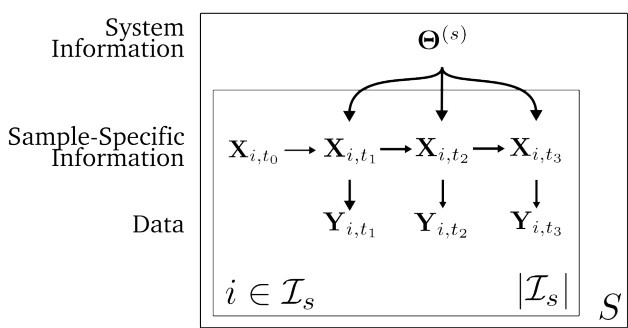

*Figure 2.* Our dynamical systems graphical model between time-series windows distinguishes transferable system information shared across time-series from non-transferable information unique to each sample (initial conditions and process noise).

## 3. PULSE Approach

In this section, we present PULSE. First, we identify which information is transferrable through a dynamical systems model that describes captures dependencies between time-series samples. Then, we introduce a practical pretraining strategy to extract the transferrable information while removing noise. Finally, we theoretically analyze conditions under which the transferrable information is recovered.

**Notation.** A physiological time-series dataset often consists of long continuous time series recordings that are segmented into shorter, fixed-length windows. Let $\mathbf{D} \in \mathbb{R}^{R \times T \times M}$ denote a dataset of $R$ recordings, each consisting of $T$ time steps and $M$ measurement channels. We use $\mathbf{D}_{r, \tau:h}$ to denote a subsequence from the $r$th recording, starting at time index $\tau$ and ending at index $h$. We can then construct a dataset of $N$ time-series windows, $\mathbf{Y} \in \mathbb{R}^{N \times W \times M}$, where each sample is defined as $\mathbf{Y}_i = \mathbf{D}_{r_i, \tau_i:\tau_i+W}$ for a window size $W$, with $r_i$ and $\tau_i$ denoting the recording and start index of the $i$th sample, respectively. A specific time-slice is denoted as $\mathbf{Y}_{i,t_1}$, where $t_1 \in \{1, \ldots, W\}$, and consecutive time-slices are defined as $t_{k+1} = t_k + 1$, where $k$ counts the steps forward from $t_1$.

### 3.1. Locating Shared Information Between Time-Series

As shown in Figure 2, we formulate a dynamical systems generative model across all time-series samples $\mathbf{Y}$, where each sample $\mathbf{Y}_i$ is produced by a latent system with parameters $\Theta_i$, and an initial condition $\mathbf{X}_{i,t_0}$. While each sample could be generated by a unique system, physiological activity is often stereotyped, with many underlying processes exhibiting consistent, repeatable patterns over time. For example, a walk cycle captured by an accelerometer displays repeated phases such as heel strike, mid-stance, and toe-off, while an ECG signal shows recurring PQRST complexes during normal sinus rhythm. As a result, different samples may share the same underlying system, and the number of unique $\Theta_i$ is generally smaller than $N$. To capture this,

we define $\mathcal{I}_s$ as the set of indices for samples generated by system $s \in \{1, \ldots, S\}$, and $\mathbf{\Theta}^{(s)}$ as the system parameters shared across all samples in $\mathcal{I}_s$. In other words, $\mathbf{\Theta}_i = \mathbf{\Theta}^{(s)}$ for all $i \in \mathcal{I}_s$. The joint distribution for this generative model is given by,

$$
\begin{aligned}
p(\mathbf{Y}, \mathbf{X}, \mathbf{\Theta}) = \prod_{s=1}^{S} \prod_{i \in \mathcal{I}_s} p(\mathbf{X}_{i,t_0}, \mathbf{\Theta}^{(s)}) \\
\left[ \prod_{k=1}^{W} p(\mathbf{Y}_{i,t_k} | \mathbf{X}_{i,t_k}) \right] \left[ \prod_{k=2}^{W} p(\mathbf{X}_{i,t_k} | \mathbf{X}_{i,t_{k-1}}, \mathbf{\Theta}^{(s)}) \right]
\end{aligned}
\tag{1}
$$

where $\mathbf{X}_{i,t_0}$ and $\mathbf{\Theta}^{(s)}$ are independent. Here, the observation density $p(\mathbf{Y}_{i,t_k} | \mathbf{X}_{i,t_k})$ is defined by the SSM measurement equation $\mathbf{Y}_{i,t_k} = g_y(\mathbf{X}_{i,t_k}) + \epsilon_{i,t_k}$ and the transition density $p(\mathbf{X}_{i,t_k} | \mathbf{X}_{i,t_{k-1}}, \mathbf{\Theta}^{(s)})$ is defined by the SSM transition equation $\mathbf{X}_{i,t_k} = g_x(\mathbf{X}_{i,t_{k-1}}, \mathbf{\Theta}^{(s)}) + \nu_{i,t_k}$, where $\epsilon_{i,t_k}$ and $\nu_{i,t_k}$ are noise terms.

The factorization in Eq. 1 reveals a hierarchy of information in $\mathbf{Y}$. One source of information comes from the system variables $\mathbf{\Theta}^{(s)}$, which govern the evolution of the time series and are shared between all $\mathbf{Y}_i$ generated by the same system. Since $\mathbf{\Theta}^{(s)}$ is shared across all samples in $\mathcal{I}_s$, it provides information that is transferable across different samples. This observation leads to the following notion of similarity between independent time-series samples.

**Definition 3.1** (Similar Time-Series). *Two time-series $\mathbf{Y}_i$ and $\mathbf{Y}_j$ are similar if they are generated by the same system parameters $\mathbf{\Theta}^{(s)}$, such that both indices satisfy $i, j \in \mathcal{I}_s$.*

Therefore, learning a representation that captures this system information produces a space where samples with similar dynamics are naturally grouped together.

Figure 2 also illustrates sample-specific sources of information that is unique to each $\mathbf{Y}_i$, such as the initial value $\mathbf{X}_{i,t_0}$, as well as observation and dynamics noise $\epsilon$ and $\nu$. In our setting, a representation should be invariant to this information, as it is not shared across $\mathbf{Y}_i$ and therefore cannot be transferred between different samples. This insight is especially relevant for physiological time series, where a signal's identity is determined by the underlying system dynamics rather than by the exact starting value, sensor noise, or transient fluctuations. Therefore, *by extracting system information and discarding sample-specific information, we obtain a representation that can relate physiological time series based on their temporal characteristics while ignoring irrelevant factors.*

### 3.2. PULSE for Recovering System Information

Our goal is to design a practical pretraining strategy that encourages an encoder to recover system information while ignoring sample-specific factors. A promising strategy is a dynamical systems *cross-reconstruction* task, where given two samples from the same system (i.e., $\mathbf{Y}_i$ and $\mathbf{Y}_j$ with

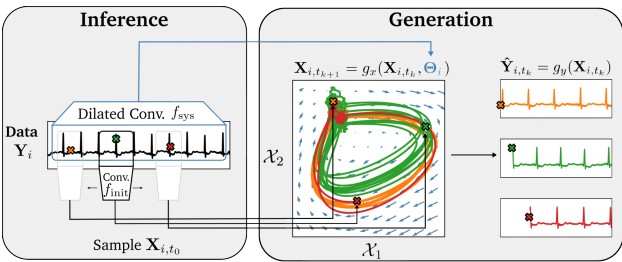

*Figure 3.* PULSE recovers system information through an inference process that uses two encoders, $f_{\text{sys}}$ to estimate shared parameters of a latent dynamical systems and $f_{\text{init}}$ to estimate sample-specific initial conditions. By requiring $\mathbf{\Theta}_i$ to support reconstruction of randomly sampled $\mathbf{X}_{i,t_0}$, it is encouraged to be invariant to sample-specific factors.

$i, j \in \mathcal{I}_s$), the system information inferred from $\mathbf{Y}_i$ is used to reconstruct an independently realized sample $\mathbf{Y}_j$. By requiring the system representation from $\mathbf{Y}_i$ to reconstruct multiple random samples of $\mathbf{Y}_j$, an encoder is encouraged to keep only the shared information between these samples. According to Eq. 1, the only shared variables are $\mathbf{\Theta}^{(s)}$, and encoding irrelevant factors may lead to poor reconstruction.

**Cross-Reconstruction with Similar Pairs.** To formalize this approach, we define an inference step and generative process given sample pairs $(\mathbf{Y}_i, \mathbf{Y}_j)$ produced by $\mathbf{\Theta}^{(s)}$ as input. For inference, we introduce two encoders that separate transferable from non-transferable dynamical systems components: a system encoder $f_{\text{sys}}$ to estimate shared system information and an initial condition encoder $f_{\text{init}}$ to estimate the sample-specific initial condition. As shown in Figure 3, $f_{\text{sys}}$ uses the dilated convolution (Yue et al., 2022) to extract information across the entire window according to $\mathbf{\Theta}_i = f_{\text{sys}}(\mathbf{Y}_i)$. In contrast, $f_{\text{init}}$ is implemented as a 2-layer CNN whose receptive field is centered around a specified time $t_0$, producing $\mathbf{X}_{j,t_0} = [f_{\text{init}}(\mathbf{Y}_j)]_{t_0}$ where $t_0 = 1$ selects the initial condition needed to reconstruct a sample from the first time step until the length of the reconstruction window $w$. For generation, we use an SSM decoder and define the cross-reconstruction objective,

$$
\begin{aligned}
& \mathcal{L}_{\text{Cross}}(\mathbf{Y}_i, \mathbf{Y}_j) \\
& = \mathbb{E}_{\mathcal{P}} \left[ \sum_{k=1}^{w} \| \mathbf{Y}_{j,t_k} - g_y(g_x(\mathbf{X}_{j,t_{k-1}}, \mathbf{\Theta}_{i,t_k})) \|^2 \right],
\end{aligned}
\tag{2}
$$

where $(\mathbf{Y}_i, \mathbf{Y}_j) \sim \mathcal{P}$ is a distribution over sample pairs $i, j \in \mathcal{I}_s$, and $g_x$ and $g_y$ are parameterized by a GRU and linear projection layer respectively. Here, we implement $\mathbf{\Theta}_i$ as the GRU input, since its hidden state evolves according to dynamics defined by input-dependent gating. Additionally, following prior dynamical systems methods (Yingzhen and Mandt, 2018), we further factorize $\mathbf{\Theta}_i$ into separate time-invariant and time-varying components to model nonstationary physiological behaviors. To do this, we decompose $\mathbf{\Theta}_i$ into a time-invariant component

$\boldsymbol{\theta}_i$, obtained via max pooling over time, and a time-varying component $\tilde{\boldsymbol{\theta}}_{i,t_k}$, obtained via a two-layer CNN over the latent dimension, and then concatenate the result at each time step to form $\boldsymbol{\Theta}_{i,t_k} = [\boldsymbol{\theta}_i, \tilde{\boldsymbol{\theta}}_{i,t_k}]$. Importantly, as shown in equation 2, $\boldsymbol{\Theta}_{i,t_k}$ does not include parameters for $g_y$ and only includes parameters for $g_x$. This design choice is motivated by the SSM formulation, where the parameters of the observation function are not considered part of the underlying dynamics, reflecting the idea that how a process is measured is separate from the dynamics of the process itself. Therefore, the resulting embedding for $\boldsymbol{\Theta}_{i,t_k}$ includes only the pooled output of the dilated convolution.

To minimize Eq. 2, $f_{\text{sys}}$ must extract shared information from $\mathbf{Y}_i$ that can explain the evolution of $\mathbf{Y}_j$. This means encoding only the underlying system variables $\boldsymbol{\Theta}^{(s)}$, since these factors remain invariant across samples from the same system. Furthermore, because $\mathcal{P}$ involves optimizing over randomly chosen pairs, $f_{\text{sys}}$ cannot rely on sample-specific factors from $\mathbf{Y}_i$, since this information will not be present in $\mathbf{Y}_j$ and may increase $\mathcal{L}_{\text{Cross}}$ if present. Thus, the $\mathcal{L}_{\text{Cross}}$ loss encourages the encoder to discard non-shared factors and focus only on shared system information.

**Cross-Reconstruction with PULSE Pseudo-Pairs.** Unfortunately, Eq. 2 requires access to system labels $\mathcal{I}_s$ from independent time-series samples, which we do not have access to in an unlabeled dataset. Therefore, we cannot directly sample from $\mathcal{P}$. Instead, we propose to construct approximately independent pseudo-pairs $(\mathbf{Y}_i, \widetilde{\mathbf{Y}}_i)$ via *system-preserving augmentations* $\widetilde{\mathbf{Y}}_i \sim \mathcal{T}(\mathbf{Y}_i)$. We define $\mathcal{T}$ as a family of transformations that maintain the underlying system's identity while providing the data variation necessary to discard sample-specific noise. For physiological time-series, preserving the system identity is crucial to ensure augmentations do not alter the clinical meaning of a signal. In our work, we use random crops for $\mathcal{T}$, since they preserve the time-series dynamics while introducing variation in the initial condition $\mathbf{X}_{i,t_0}$. This aligns with our generative model in Fig. 2 and encodes the assumption that the underlying physiological state is often independent of the specific moment recording begins. By requiring $\boldsymbol{\Theta}_i$ to support accurate reconstruction across random crops, the system encoder must learn to recover system dynamics that is invariant to the specific starting state. This leads to the PULSE objective,

$$\mathcal{L}_{\text{PULSE}}(\mathbf{Y}_i) =$$
$$\mathbb{E}_{\widetilde{\mathbf{Y}}_{i,t_k} \sim \mathcal{T}} \left[ \sum_{k=1}^{w} \|\widetilde{\mathbf{Y}}_{i,t_k} - g_y(g_x(\mathbf{X}_{i,t_{k-1}}, \boldsymbol{\Theta}_{i,t_k}))\|^2 \right], \quad (3)$$

where $\mathbf{X}_{i,t_0} = [f_{\text{init}}(\mathbf{Y}_i)]_{t_0}$ and $\boldsymbol{\Theta}_i = f_{\text{sys}}(\mathbf{Y}_i)$. Here, the system-preserving augmentation $\widetilde{\mathbf{Y}}_{i,t_k} \sim \mathcal{T}(\mathbf{Y}_i)$ is a random crop of length $w$, where the initial time of the cropped window is $t_0 \sim \text{Uniform}(1, T - w)$. We find that using

multiple $\widetilde{\mathbf{Y}}_i$'s to estimate the expectation can improve performance, and in our experiments we use up to four.

**Regularizing Time-Varying System Variables.** Since both encoders observe the same $\mathbf{Y}_i$, and the system representation includes $\boldsymbol{\theta}_{i,t_k}$ derived from that same input, it is possible that the encoder learns a trivial representation of the dynamics by simply copying local signal values into these $\boldsymbol{\theta}_{i,t_k}$. We mitigate this scenario with the following two strategies that limit the expressivity of $\boldsymbol{\theta}_{i,t_k}$. First, we reduce the dimensionality of the $\boldsymbol{\theta}_{i,t_k}$ to a single dimension similar to prior works (Yingzhen and Mandt, 2018). This ensures that $\boldsymbol{\theta}_{i,t_k}$ alone does not have sufficient capacity to represent the full diversity of initial conditions in the data. Second, we limit how quickly $\boldsymbol{\theta}_{i,t_k}$ can change over time by sharing max pooled values across the time dimension between consecutive timesteps.

### 3.3. Provable Recovery of System Information

We now provide a theoretical analysis of our framework and identify conditions where cross-reconstruction provably recovers system information. Our strategy builds on prior work showing that MAE pretraining implicitly recovers information from the minimal set of latent variables shared[2] $\mathcal{C}$ between masked and unmasked regions in a hierarchical data-generating process (Kong and Zhang, 2023). By viewing cross-reconstruction as an MAE task under a specific masking strategy, we can extend this theory to characterize the type of information recovered in our time-series generative model under different masks.

Cross-reconstruction can be viewed as an MAE task by treating the pair $(\mathbf{Y}_i, \mathbf{Y}_j)$ as a single joint input with a pair of masking variables $(\mathbf{m}_i, \mathbf{m}_j)$, where each $\mathbf{m}_i \in \{0,1\}^{W \times M}$ indicates for every element $(w, m)$ whether it is observed ($m_{i,w,m} = 1$) or masked ($m_{i,w,m} = 0$). In this view, Eq. 2 corresponds to setting $\mathbf{m}_{i,1:W,1:M} = 1$ to retain $\mathbf{Y}_i$ as input, while fully masking the other sample $\mathbf{m}_{j,1:W,1:M} = 0$, so that $\mathbf{Y}_j$ is removed and can serve as the reconstruction target. The effect of this masking strategy on the type of information recovered can then be characterized given our generative model in Eq. 1. To make this precise, we outline our assumptions (further detailed in Appendix A) on the generative process.

**Assumption 1.** *(Data-Generating Process). The process in Equation* (1) *satisfies the following conditions: (i) the fully factorized generative model is a directed acyclic graph (DAG) ; and (ii) each function $g_k$ is invertible.*

Given this data-generating process and the cross-reconstruction masking scheme described above, we present our theory, which identify the $\mathcal{C}$ between masked

---

[2]Our minimal set of shared latent variables $\mathcal{C}$ is defined in Theorem 1 of (Kong and Zhang, 2023) as $\mathbf{c}$.

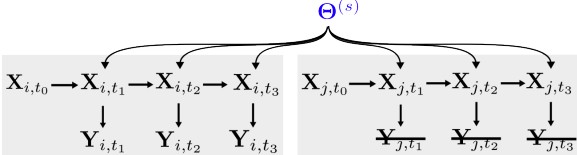     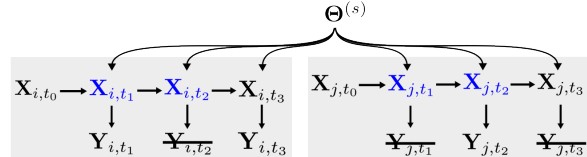

A) Positive Mask: System Information Recovered     B) Negative Mask: Sample-Specific Information Recovered

*Figure 4.* We illustrate how different masking strategies can change the source of information recovered in our data-generating process for sample pairs $(\mathbf{Y}_i, \mathbf{Y}_j)$ where $i, j \in \mathcal{I}_s$. $\mathbf{Y}$ marks an observable that is removed from the input and used as a reconstruction target. **Blue** highlights $\mathcal{C}$, representing the information that is recovered during pretraining. Theorem 1 predicts that $\mathcal{C} = \{\mathbf{\Theta}^{(s)}\}$ only when information from one sample is fully removed. Gray boxes group latent variables that are specific to each time-series sample.

*Table 1.* Our synthetic results confirm Theorem 1's predictions: The positive oracle improves classification accuracy over PULSE pretraining without labels, while the negative oracle reduces performance relative to the positive model and, at $\sigma = 5$, performance even falls below the practical PULSE algorithm. PULSE is the most effective practical algorithm, being the best at distinguishing parameter changes across all $\sigma$. **Black** is the best practical algorithm (pretrained without labels), while **blue** is the best oracle (pretrained with labels) when it exceeds all practical methods. Fig. 4 illustrates the masking strategy for positive and negative oracle.

| Noise $\sigma$ | SimCLR | TS2Vec | REBAR | PatchTST | TimeMAE | LFADS | DSVAE | PULSE | PULSE Oracle Positive | PULSE Oracle Negative |
|---|---|---|---|---|---|---|---|---|---|---|
| 0 | 93.08 | 98.68 | 98.90 | 77.59 | 96.29 | 99.06 | 98.56 | **99.58** | 99.29 | 98.86 |
| 1 | 83.10 | 93.07 | 93.36 | 50.36 | 93.42 | 93.02 | 90.84 | **96.09** | **97.26** | 96.66 |
| 3 | 70.05 | 79.78 | 79.36 | 39.88 | 75.08 | 79.03 | 76.70 | **83.42** | **89.00** | 84.62 |
| 5 | 62.29 | 73.67 | 72.37 | 37.82 | 66.63 | 71.33 | 69.90 | **77.34** | **82.65** | 76.90 |

and unmasked regions. This $\mathcal{C}$ corresponds to the information that is implicitly recovered during MAE pretraining (Kong and Zhang, 2023).

**Theorem 1.** *Given two time series $\mathbf{Y}_i$ and $\mathbf{Y}_j$ independently sampled from the same system (i.e., $\mathbf{\Theta}_i = \mathbf{\Theta}_j = \mathbf{\Theta}^{(s)}$) under the generative process defined by Eq. 1 and Assumption 1, the minimal set of latent variables shared is the system parameters $\mathbf{\Theta}^{(s)}$ if and only if all observables from one series is fully masked (i.e., $\mathbf{m}_{i,1:W,1:M} = 0$ and $\mathbf{m}_{j,1:W,1:M} = 1$).*

A proof is provided in Appendix A. Theorem 1 states that system information is recovered during a masked reconstruction task when an entire time series is removed from the input pair, as this masking scheme uniquely ensures that the $\mathcal{C}$ connecting the masked and unmasked regions contains the system parameters. Importantly, this reconstruction task with whole-sample masking corresponds exactly to the pretraining objective $\mathcal{L}_{\mathrm{Cross}}$. Moreover, as shown in Fig. 4, this theory predicts that when a time series contains both masked and unmasked regions, $\mathcal{C}$ necessarily includes the state variables $\mathbf{X}$, causing the recovered information to confound sample-specific and system information. Since $\mathcal{L}_{\mathrm{PULSE}}$ can be viewed as an approximation of $\mathcal{L}_{\mathrm{Cross}}$ that uses pseudo-pairs to simulate independent samples, this theory offers an explanation for how $\mathcal{L}_{\mathrm{PULSE}}$ can recovers system information. Appendix **??** provides additional discussion on the properties of independent samples that PULSE approximates.

## 4. Synthetic Dynamical Systems Experiments

**Set up.** We investigate Theorem 1 in a setting where Assumption 1 may not completely hold. Specifically, we consider a synthetic dynamical systems experiment, where $g_x$ cannot be practically inverted due to system chaos. The goal of this task is to learn a representation space that can distinguish between parameter settings while remaining robust to increasing levels of dynamical system noise. This task is motivated by real-world scenarios in which parameter changes in the underlying system may correspond to shifts in physiological state, and effective representations must reliably capture these shifts. Time series data are generated from three stochastic differential equations (Lorenz, Thomas, and Hindmarsh-Rose) across a grid of parameters in bifurcation regions and noise levels $\sigma = \{0, 1, 3, 5\}$. Datasets are then constructed by randomly selecting five parameter settings to form a five-class classification problem, with trials split into 70:15:15 for train, validation, and test sets and subsequently segmented into windows where $W = 100$. For each $\sigma$, linear probe results are averaged over 10 random dataset samples and model initializations per system. Full dataset details are provided in Appendix B.

**Results.** Table 1 shows that PULSE consistently achieves the highest classification accuracy among all practical baselines (described in Section 5 and Appendix C), even as $\sigma$ increases. This demonstrates that PULSE pretraining on pseudo-pairs $(\mathbf{Y}_i, \tilde{\mathbf{Y}}_i)$ is more robust to noise and can extract class-discriminative features that reveal changes in system parameters. Furthermore, we validate Theorem 1

by considering a PULSE Oracle where label information is used to identify true $(\mathbf{Y}_i, \mathbf{Y}_j)$ pairs to construct the positive and negative set ups illustrated in Figure 4. Specifically, the positive model leverages labels to select pairs in $\mathcal{L}_{\mathrm{Cross}}$, whereas the negative model applies random temporal masking to each pair before inputting both into $f_{\mathrm{init}}$ and $f_{\mathrm{sys}}$. According to Theorem 1, the negative oracle captures sample-specific information that is uninformative for classifying system parameters, so the positive oracle is expected to consistently outperform it. Our results in Table 1 confirm this prediction with positive oracle outperforming all practical algorithms and negative oracle consistently underperforming the positive oracle.

## 5. Real Physiological Data Experiments

We evaluate PULSE against representative SSL baselines across several real-world datasets and downstream tasks.

**Data.** We consider 4 commonly used physiological time-series datasets from distinct sensor domains, each consisting of long trials with time-varying classification labels. We use Human Activity Recognition (HAR) (Reyes-Ortiz et al., 2015), where human activity is estimated from accelerometer and gyroscope signals; PPG (Schmidt et al., 2018), where optical blood volume signals are used to estimate stress levels; ECG (Moody, 1983), where the heart's electrical activity is used to detect rhythm abnormalities; and EEG (Kemp et al., 2000), where the brain's electrical activity is used to estimate sleep stages. A detailed description of these datasets is provided in Appendix D.

**Baselines.** We benchmark performance against a representative set of SSL pretraining approaches, including three CL methods: SimCLR (Chen et al., 2020), TS2Vec (Yue et al., 2022), and REBAR (Xu et al., 2024), two SVAE models: LFADS (Sedler and Pandarinath, 2023; Sussillo et al., 2016) and DSVAE (Yingzhen and Mandt, 2018), and two MAE approaches: TimeMAE (Cheng et al., 2026) and PatchTST (Nie et al., 2022). To evaluate the pretraining objective, we use the same dilated convolution encoder architecture (Yue et al., 2022) across all CL and SVAE experiments, ensuring that performance differences reflect the quality of the pretraining rather than differences in architectures. For MAE baselines, we retain the original transformer encoders, since changing it reduces performance. Appendix C provides additional details on baselines.

### 5.1. Linear Probe Evaluation

To assess the ability of pretraining to learn class-discriminative features, we train a linear probe (logistic regression) on the frozen embeddings from each pretrained model to predict the ground truth physiological class labels from each dataset. We measure performance by re-porting Accuracy, AUROC, and AUPRC averaged over 5 random model initializations over a single pre-defined training-val-test split (details in Appendix D and full cross-validation experiments in Appendix E). Table 2 shows that PULSE pretraining achieves strong linear probe performance across all four datasets, performing competitively on HAR and achieving the highest overall scores on PPG, ECG, and EEG. Notably, PULSE substantially outperforms LFADS and DSVAE on ECG and PPG, highlighting that explicitly removing noise provides an clear advantage over SVAE objectives that do not distinguish between noise and signal. Moreover, the improvements over Sim-CLR, TS2Vec, and REBAR show that CL's sensitivity to false positive pairs may limit its effectiveness across diverse sensor domains. Finally, the performance increase over PatchTST and TimeMAE demonstrates that designing a generative task to explicitly extract system information produces representations that better distinguish physiological states than those learned by standard masked modeling, which does not leverage this structure.

### 5.2. Semi Supervised Evaluation

Next, we evaluate the label efficiency of the pretrained representations using a semi-supervised classification task on the pretrained frozen embeddings. For each pretrained model, we train a linear probe on 1% and 5% of the ground truth labels and apply Laplace smoothing so that all downstream classes are represented. Reported accuracies are averaged over five random label subsets for each of the five model initializations from Section 5.1, for a total of 25 seeds. We also include a supervised baseline to estimate performance achievable without pretraining. Table 3 shows that PULSE pretraining consistently outperforms the baselines across all four sensor domains, achieving the highest scores among SSL methods across all datasets. Interestingly, PULSE's HAR representation is more label-efficient, achieving strong performance with very few labels despite lower linear probe scores on the full labeled dataset, suggesting that it captures key class-discriminative features more efficiently. PULSE also outperforms most supervised baselines, highlighting the advantage of its pretrained representations in limited data scenarios.

### 5.3. Transfer Learning Evaluation

We investigate in-domain transfer learning for classification in two scenarios from (Zhang et al., 2022). This task is motivated by real-world applications where we want to transfer knowledge between datasets collected from similar sensors. In the first scenario, a model is pretrained on EEG (Kemp et al., 2000) and fine-tuned on the Epilepsy dataset (Andrzejak et al., 2001). In the second, a model is pretrained on HAR (Reyes-Ortiz et al., 2015) and fine-tuned on Gesture (Liu et al., 2009). Our setup follows (Zhang et al., 2022), where we attach a 2-layer MLP head and

*Table 2.* Linear Probe Classification. PULSE achieves the best results on PPG, ECG, and EEG. Note that while HAR scores for PULSE are lower than SOTA in this experiment, this representation leads to improved performance in Tables 3 and 4.

| | Metric | SimCLR | TS2Vec | REBAR | PatchTST | TimeMAE | LFADS | DSVAE | PULSE |
|---|---|---|---|---|---|---|---|---|---|
| **HAR** | Accuracy ↑ | 94.65 | 93.24 | **95.35** | 83.04 | 92.25 | 93.55 | 93.55 | 93.27 |
| | AUROC ↑ | 99.38 | 99.31 | **99.65** | 97.44 | 99.14 | 99.49 | 99.36 | 99.42 |
| | AUPRC ↑ | 97.63 | 97.66 | **98.91** | 89.90 | 97.05 | 98.29 | 97.69 | 98.10 |
| **PPG** | Accuracy ↑ | 34.48 | 40.23 | 41.38 | 59.78 | 61.35 | 52.81 | 58.65 | **64.27** |
| | AUROC ↑ | 61.19 | 64.28 | 69.77 | 71.08 | 78.08 | 71.10 | 76.78 | **80.29** |
| | AUPRC ↑ | 36.08 | 39.59 | 44.57 | 52.91 | 56.74 | 49.59 | 55.38 | **59.89** |
| **ECG** | Accuracy ↑ | 69.92 | 76.12 | 81.54 | 64.40 | 69.80 | 61.84 | 70.42 | **87.41** |
| | AUROC ↑ | 82.54 | 86.56 | 91.46 | 70.96 | 76.61 | 71.69 | 82.88 | **94.93** |
| | AUPRC ↑ | 80.63 | 85.16 | 89.85 | 68.16 | 76.62 | 69.21 | 81.31 | **94.75** |
| **EEG** | Accuracy ↑ | 66.38 | 83.76 | 83.71 | 80.62 | 83.83 | 82.43 | 84.25 | **85.56** |
| | AUROC ↑ | 85.45 | 94.99 | 95.08 | 93.55 | 95.09 | 94.49 | 95.42 | **96.17** |
| | AUPRC ↑ | 50.95 | 70.22 | 70.77 | 67.37 | 73.22 | 68.55 | 72.25 | **73.82** |

*Table 3.* Semi-supervised classification accuracy for 1% and 5% of labels averaged over 25 random seeds. Higher score is better. PULSE outperforms all SSL baselines and most supervised baselines.

| Dataset | | Supervised | SimCLR | TS2Vec | REBAR | PatchTST | TimeMAE | LFADS | DSVAE | PULSE |
|---|---|---|---|---|---|---|---|---|---|---|
| | HAR | 78.39 | 71.74 | 80.57 | 81.10 | 33.27 | 80.79 | 80.97 | 79.94 | **84.74** |
| 1 % | ECG | 45.46 | 63.83 | 62.77 | 67.56 | 57.68 | 65.15 | 57.34 | 67.60 | **84.77** |
| | PPG | 34.38 | 30.34 | 31.98 | 32.33 | 41.74 | 40.45 | 40.22 | 41.28 | **42.97** |
| | EEG | 77.76 | 56.72 | 77.39 | 77.19 | 63.45 | 70.55 | 74.19 | 78.40 | **80.69** |
| | HAR | 92.34 | 85.01 | 91.76 | 91.04 | 54.79 | 91.55 | 91.48 | 90.72 | **93.14** |
| 5 % | ECG | 69.20 | 65.00 | 63.97 | 70.12 | 60.73 | 68.73 | 60.04 | 67.66 | **84.23** |
| | PPG | 42.47 | 30.62 | 33.13 | 38.25 | 49.48 | 49.53 | 45.35 | 51.15 | **53.39** |
| | EEG | **84.90** | 61.98 | 77.09 | 76.75 | 73.48 | 77.50 | 75.19 | 78.17 | 80.45 |

*Table 4.* In-Domain Transfer Learning.

| | EEG → Epilepsy | | HAR → Gesture | |
|---|---|---|---|---|
| | ACC | AUROC | ACC | AUROC |
| SimCLR | 93.52 | 97.52 | 78.83 | 93.80 |
| TS2Vec | 93.95 | 95.87 | 77.67 | 95.45 |
| REBAR | 95.27 | 98.33 | 78.17 | 95.54 |
| PatchTST | 95.03 | 98.04 | 77.00 | 95.21 |
| LFADS | 94.71 | 98.01 | 78.50 | 95.42 |
| DSVAE | 94.97 | 98.17 | 78.00 | 95.34 |
| PULSE | **95.82** | **98.51** | **83.67** | **97.20** |

*Table 5.* Ablation study. Relative effect ($\Delta$) of removing model components on linear probe accuracy.

| | HAR | ECG | PPG | EEG | Avg. |
|---|---|---|---|---|---|
| PULSE | **93.27** | **87.41** | **64.27** | **85.56** | - |
| w/o TV-Params $\tilde{\theta}_{i,t_k}$ | -1.73 | -6.83 | -9.22 | -12.7 | -7.62 |
| Shared $f_{sys}$ and $f_{init}$ | -1.02 | -1.75 | -1.58 | -0.54 | -1.22 |
| Direct Recon. (w/o Random Crop) | -1.16 | -7.73 | -15.96 | -0.81 | -6.42 |
| Cross-Recon. w Random Pairs | -2.42 | -22.99 | -6.96 | -6.35 | -9.68 |

fine-tune for 40 epochs using the Adam optimizer with a learning rate of 0.0003. A detailed description of the fine-tuning datasets is provided in Appendix D. Table 4 reports accuracy and AUROC averaged over five random seeds for both model initialization and fine-tuning. In both scenarios, PULSE consistently achieves the highest transfer performance, with a substantial gain on the HAR-to-Gesture task. This demonstrates that pretraining designed to explicitly prioritize system information can produce features that transfer effectively to related downstream tasks.

### 5.4. Ablation Study

Table 5 shows the effect of ablating various PULSE components on the linear probe accuracy across all datasets, reported as $\Delta = \text{Acc}_{\text{Ablated}} - \text{Acc}_{\text{PULSE}}$. In *w/o TV-Params*

$\tilde{\theta}_{i,t_k}$, we remove time-varying parameters and retain only time-invariant ones, i.e., $\boldsymbol{\Theta}_i = \boldsymbol{\theta}_i$. This results in an average accuracy drop of 7.6%, underscoring the importance of modeling non-stationary dynamics in physiological time series. In *Shared $f_{sys}$ and $f_{init}$*, we estimate $\mathbf{X}_{i,t_k}$ from the output of $f_{sys}$ such that $\mathbf{X}_{i,t_k} = [f_{init}(f_{sys}(\mathbf{Y}_i))]_{t_k}$, rather than using two separate encoders. This results in a 1.22% drop, suggesting that explicitly separating transferrable and non-transferrable information can improve time-series representation learning. In *w/o random crops*, we fix $t_0 = 1$ for all samples during initial-condition inference. This effectively reduces the method to direct reconstruction, as the input–output pairs become fixed rather than randomly sampled. This leads to a 6.4% drop in average accuracy, showing that training the system representation with system-preserving augmentations is important and that our pseudo-pair strategy can effectively construct similar time-series pairs without in unlabeled datasets. In *Cross Reconstruc-*

*tion with Random Pairs*, we form input–output pairs by randomly sampling segments without regard for the underlying dynamics of each time-series sample. This ablation has the largest effect, decreasing the average performance by -9.68%. This emphasizes that effective cross-reconstruction requires identifying similar time-series and further highlighting the effectiveness of our pseudo-pair strategy in achieving this.

## 6. Conclusion

In this work, we introduced PULSE, an SSL pretraining framework that simultaneously preserves dynamical systems information while selectively filtering out sample-specific noise. Through our novel cross-reconstruction strategy, PULSE combines the structure of VAE models with the transferability of modern SSL methods. We hope this work inspires future research into better formulations of the underlying generative model to obtain improved representations of more complex physiological phenomena.

## Impact Statement

This paper presents work whose goal is to advance the field of machine learning. There are many potential societal consequences of our work, none of which we feel must be specifically highlighted here.

## Acknowledgements

This work was funded by the James S. McDonnell Foundation (grant no. 22002039), the National Institutes of Health (grant nos. 2T32EB025816 and P41EB028242), the Julian T. Hightower Chair at Georgia Tech, and the National Science Foundation Graduate Research Fellowship (grant no. DGE-2039655). The authors are part of the Georgia Tech/Emory NIH/NIBIB Training Program in Computational Neural Engineering (T32EB025816). The content is solely the responsibility of the authors and does not necessarily represent the official views of the NIH or NSF.

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

# Appendix

## A. Proof for Theorem 1

**Assumption 1.** *(Data-Generating Process). The process in Equation* (1) *satisfies the following conditions: (i) the fully factorized generative model is a directed acyclic graph (DAG) ; and (ii) each function $g_k$ is invertible.*

**Assumption Interpretation.** Part (i) ensures that our theory applies to complex systems with elaborate parameter factorizations, as long as they remain acyclic. Part (ii) guarantees that no information is lost during the generative process and is adopted from prior work on identifiable deep generative models (Locatello et al., 2020; Von Kügelgen et al., 2021).

**Theorem 1.** *Given two time series $\mathbf{Y}_i$ and $\mathbf{Y}_j$ independently sampled from the same system (i.e., $\Theta_i = \Theta_j = \Theta^{(s)}$) under the generative process defined by Eq. 1 and Assumption 1, the minimal set of latent variables shared is the system parameters $\Theta^{(s)}$ if and only if all observables from one series is fully masked (i.e., $\mathbf{m}_{i,1:W,1:M} = 0$ and $\mathbf{m}_{j,1:W,1:M} = 1$).*

*Proof.* Our proof relies on Theorem 1 from (Kong and Zhang, 2023), which establishes that the minimal set of shared information $\mathbf{c}$ in a hierarchical DAG is unique and can be identified using Algorithm 1 presented in their work. We describe the generative model that is considered and summarize Algorithm 1 before applying it in the proof.

**Two-sample Generative Model.** We consider the cross-reconstruction setting with a pair of time series $(\mathbf{Y}_i, \mathbf{Y}_j)$ for $i, j \in \mathcal{I}_s$, generated from the same system such that $\Theta_i = \Theta_j = \Theta^{(s)}$. When considering only two samples, the joint distribution in Eq. 1 reduces to the following form,

$$p(\mathbf{Y}_i, \mathbf{Y}_j, \mathbf{X}_i, \mathbf{X}_j, \Theta^{(s)}) = p(\Theta^{(s)}) \prod_{n=\{i,j\}} p(\mathbf{X}_{n,t_0})$$
$$\left[ \prod_{k=1}^{W} p(\mathbf{Y}_{n,t_k} | \mathbf{X}_{n,t_k}) \right] \left[ \prod_{k=2}^{W} p(\mathbf{X}_{n,t_k} | \mathbf{X}_{n,t_{k-1}}, \Theta^{(s)}) \right].$$
$$(4)$$

Moreover, we denote masked and unmasked observables respectively as,

$$\mathbf{Y_m} = \bigcup_{k=\{i,j\}} \{\mathbf{Y}_{k,t} \mid \mathbf{m}_{k,t} = 0\} \quad \text{and}$$
$$\mathbf{Y_{m_c}} = \bigcup_{k=\{i,j\}} \{\mathbf{Y}_{k,t} \mid \mathbf{m}_{k,t} = 1\}.$$

In words, $\mathbf{Y_m}$ and $\mathbf{Y}_{m_c}$ are the sets of masked and unmasked time-points across both samples, respectively.

**Algorithm 1 from (Kong and Zhang, 2023).** This approach uses a two-stage procedure for identifying the minimal set of latent variables $\mathcal{C}$ shared between masked and unmasked observables in a hierarchical graphical model. In the first stage, called the *selection stage*, all latent variables that are ancestors of both $\mathbf{Y_m}$ and $\mathbf{Y}_{m_c}$ are located. This is done by collecting all parents of the masked observables and adding those that are also ancestors of the unmasked observables to $\mathcal{C}$. The resulting set $\mathcal{C}$ contains all latent variables shared between $\mathbf{Y_m}$ and $\mathbf{Y}_{m_c}$. In the second stage, called the *pruning stage*, each element of $\mathcal{C}$ is checked to ensure that no other element in $\mathcal{C}$ lies on the directed path between it and $\mathbf{Y}_{m_c}$. If such a descendant exists, the ancestor node is removed from $\mathcal{C}$. To summarize, given a hierarchical graphical model with masked and unmasked observables $\mathbf{Y_m}$ and $\mathbf{Y}_{m_c}$, this algorithm returns $\mathcal{C}$, the set of variables with lowest dimension that block all paths between $\mathbf{Y_m}$ and $\mathbf{Y}_{m_c}$.

Now, we prove each direction of the if-and-only-if condition separately.

**If a full sample is masked, then $\mathcal{C} = \Theta^{(s)}$.** Without loss of generality, we define full-sample mask as $\mathbf{m}_i = 0$ and $\mathbf{m}_j = 1$ for $i, j \in \mathcal{I}_s$. In this case, any element in $\mathcal{C}$ must be a common parent of both samples. As illustrated in Fig. 4A, the only parent node that is shared between $\mathbf{Y}_i$ and $\mathbf{Y}_j$ is $\Theta^{(s)}$, and thus the only set of variables shared between masked and unmasked regions is the system variables. When Algorithm 1 from (Kong and Zhang, 2023) is applied to the graphical model in Eq. 4 under the full-sample masking scheme, it recovers $\mathcal{C} = \{\Theta^{(s)}\}$. In the selection stage, $\Theta^{(s)}$ is the only parent node connecting both samples $\mathbf{Y}_i$ and $\mathbf{Y}_j$. In the pruning stage, nothing is removed since $\mathcal{C}$ contains only a single element, implying no additional shared latent variables exist as children of $\Theta^{(s)}$. Therefore, under full-sample masking, the minimal set of shared latent variables includes only the shared system parameters $\Theta^{(s)}$.

**If $\mathbf{c} = \Theta^{(s)}$, then a full sample is masked.** We prove the statement via its contrapositive: if a sample is not fully masked (i.e. contains both masked and unmasked observables), then the minimal set of shared latent variables cannot consist solely of the system parameters.

Intuitively, as shown in Fig. 4B, when a sample contains both masked and unmasked time-points, there is always a latent state variable $\mathbf{X}$ that serves as the parent node connecting $\mathbf{Y_m}$ and $\mathbf{Y}_{m_c}$. To formalize this, we define a *subsequence mask* as a consecutive region of masked observables, i.e., $\mathbf{m}_{i,t_0:t_1} = 0$, where $t_0, t_1 \in 1, \ldots, T$, $t_0 \leq t_1$, and $t_1 - t_0 < T$ for a partial mask in a sample of length $T$. Thus, $t_0 = t_1$ corresponds to masking a single time point at the index $t_0$.

There are three possible types of masked subsequence re-

gions: (1) a mask bordering the left edge, or the beginning of the sample, (2) a mask bordering the right edge, or the end of the sample, and (3) a mask in the middle of the sample that is bordered on the left and right by unmasked regions. We apply Algorithm 1 in (Kong and Zhang, 2023) to each case to determine what minimal set of shared latent variables are recovered. We use $\mathcal{C}_{\text{subseq}}$ to denote the minimal set of shared latent variables between masked and unmasked regions induced by a subsequence mask.

*Case 1 (Left Edge):* When the mask borders the left edge of sample $\mathbf{Y}_i$, the masked region satisfies $t_0 = 1$ and $t_1 < T$. During the selection stage, Algorithm 1 retrieves $\mathcal{C}_{\text{subseq}} = \{\boldsymbol{\Theta}^{(s)}, \mathbf{X}_{i,t_1}\}$, since these nodes are ancestors of both masked and unmasked regions. In the pruning stage, $\boldsymbol{\Theta}^{(s)}$ is removed because $\mathbf{X}_{i,t_1}$ lies on the directed path between $\boldsymbol{\Theta}^{(s)}$ and the unmasked region. Therefore, $\mathcal{C}_{\text{subseq}} = \{\mathbf{X}_{i,t_1}\}$. Intuitively, the latent variable $\mathbf{X}_{i,t_1}$ serves as the minimal parent connecting the masked and unmasked unmasked regions, $\mathbf{Y}_{i,1:t_1}$ and $\mathbf{Y}_{i,t_1:T}$ respectively.

*Case 2 (Right Edge):* The right edge follows a similar analysis, where the masked region satisfies $t_0 > 1$ and $t_1 = T$. This results in $\mathcal{C}_{\text{subseq}} = \{\boldsymbol{\Theta}^{(s)}, \mathbf{X}_{i,t_0}\}$ after the selection stage, and $\mathcal{C}_{\text{subseq}} = \{\mathbf{X}_{i,t_0-1}\}$ after the pruning stage. Thus, the latent variable above the $\mathbf{X}_{i,t_0-1}$ serves as the lowest-level parent connecting unmasked and masked regions, $\mathbf{Y}_{i,1:t_0}$ and $\mathbf{Y}_{i,t_0:T}$ respectively.

*Case 3 (Middle):* When $t_0 > 1$ and $t_1 < T$, the masked region is bordered on both the left and right by unmasked variables. During the selection stage, $\mathcal{C}_{\text{subseq}} = \{\mathbf{X}_{i,t_0-1}, \mathbf{X}_{i,t_1}, \boldsymbol{\Theta}^{(s)}\}$, and after pruning, $\mathcal{C}_{\text{subseq}} = \{\mathbf{X}_{i,t_0-1}, \mathbf{X}_{i,t_1}\}$, since the latent state variables lie on the directed paths from the system variables to the unmasked regions. Therefore, there are two minimal parent nodes: one at the left boundary, $\mathbf{X}_{i,t_0-1}$, above the unmasked variable $\mathbf{Y}_{i,t_0-1}$, and one at the right boundary, $\mathbf{X}_{i,t_1}$, above the masked variable $\mathbf{Y}_{i,t_1}$.

Putting these results together, the minimal shared latent variables induced by a subsequence mask is given by,

$$
\mathcal{C}_{\text{subseq}} = \begin{cases} \{\mathbf{X}_{i,t_1}\}, & \text{if } t_0 = 1 \text{ and } t_1 < T, \\ \{\mathbf{X}_{i,t_0-1}\}, & \text{if } t_0 > 1 \text{ and } t_1 = T, \\ \{\mathbf{X}_{i,t_0-1}, \mathbf{X}_{i,t_1}\}, & \text{if } t_0 > 1 \text{ and } t_1 < T. \end{cases}
\tag{5}
$$

We can extend this result to arbitrary masks, since any masking configuration over timepoints can be expressed as a union of subsequence masks, $\mathbf{m}_i = \bigcup_k \mathbf{m}_{i,t_0^{(k)}:t_1^{(k)}}$. To enforce partial masking, we require that $\mathbf{m}_{i,t} = 0$ for some $t$ and $\mathbf{m}_{i,t'} = 1$ for some $t' \neq t$. Consequently, for arbitrary masks, the minimal shared latent set is the union over the minimal sets for each subsequence mask $\mathcal{C} = \bigcup_k \mathcal{C}_{\text{subseq}}^{(k)}$. Since each $\mathcal{C}_{\text{subseq}}$ contains only latent state variables, the union $\mathcal{C}$ does not include the system variables. Therefore, we have shown that under partial masking, the minimal set of shared latent variables does not include the system parameters.

To summarize, applying Algorithm 1 from (Kong and Zhang, 2023) under a full-sample masking scheme yields $\mathcal{C} = \{\boldsymbol{\Theta}^{(s)}\}$, since it is the only variable shared between samples in our graphical model. In contrast, under partial masking, the minimal shared latent set includes only the latent state variables at the boundaries of the masked subsequences, and never the system parameters. This follows directly from our hierarchical structure, as the $\mathbf{X}$ variables always lie on the directed path from $\boldsymbol{\Theta}^{(s)}$ to $\mathbf{Y}_{\mathbf{m}_c}$.

Thus, under our two-sample generative model and applying Algorithm 1 from (Kong and Zhang, 2023), the system parameters appear as the minimal shared latent variables if and only if an entire sample is masked.

$\square$

## B. Synthetic Dataset Description

Synthetic time-series trials $\mathbf{y} \in \mathbb{R}^{T \times M}$ of length $T$ and $M$ measurement dimensions are generated by numerically integrating the Stratonovich SDE,

$$
\mathrm{d}\mathbf{y}_t = f(\mathbf{y}_t)\mathrm{d}t + \tilde{\sigma}\mathrm{d}\mathbf{B}_t,
\tag{6}
$$

with a fixed step size $\mathrm{d}t = 10^{-3}$, where $f(\cdot)$ is the dynamics function, $\mathbf{B}_t$ is multidimensional Brownian noise. To ensure that the noise levels are comparable across different systems, we set the diffusion scale as $\tilde{\sigma} = \sigma \, \text{RMS}(\mathbf{Y})$ where $\sigma$ is a dimensionless noise level and $\text{RMS}(\mathbf{Y})$ is the root-mean-square amplitude of the noiseless time-series and is estimated empirically through samples $\mathbf{Y}$ from the system. We consider the following noise levels $\sigma = \{0, 1, 3, 5\}$ and integrate eq. 6 using `torchsde` (Li et al., 2020; Kidger et al., 2021).

We generate time series from three dynamical systems: Lorenz, Thomas, and Hindmarsh-Rose. These define strange attractor that produces bounded yet nontrivial dynamics. Importantly, for certain parameter regimes, these systems undergo bifurcations, where changes to parameters induce qualitative changes in the dynamics, thereby altering the statistical properties of the resulting time series. We select systems whose behavior is sensitive to parameter changes, as physiological time series are also generated by nonlinear dynamical systems that are highly sensitive to changes in their underlying parameters. For example, the bursting behavior of a neuron can be triggered or suppressed depending on the inputs it receives (Hindmarsh and Rose, 1984; Kim and Lim, 2019). For each parameter setting and noise level, we generate 20 long time series with $T = 10^5$ time-steps from random initial conditions

$\mathbf{Y}_{i,0} \sim \mathcal{N}(0, I)$ and discard the first 200 steps as a burn-in period to ensure convergence to the attractor manifold. Below, we detail the parameters that we consider for each system.

**Lorenz (Lorenz, 1963).** Although originally derived from atmospheric convection, the Lorenz attractor serves as a conceptual tool for studying physiological dynamics. It exhibits bounded, irregular, and parameter-sensitive behavior, features shared by many biological systems such as heart rate variability (Billman, 2011) and neural activity (Chen et al., 2024; Mudrik et al., 2024; Sussillo et al., 2016). This is the canonical 3D nonlinear attractor used to study chaotic behavior in dynamical systems, with a state-space trajectory that resembles butterfly wings. For this system, $M = 3$, and the dynamics are given by,

$$\frac{d\mathbf{y}}{dt} = \begin{bmatrix} s(y_2 - y_1) \\ y_1(\rho - y_3) - y_2 \\ y_1 y_2 - \beta y_3 \end{bmatrix} \tag{7}$$

where $\mathbf{y} = [y_1, y_2, y_3]^\top$. Following prior work (Sparrow, 2012; Kamiya et al., 2024), we fix $\beta = 8/3$ and $s = 28$, and sweep $\rho$ across the following 10 values: $\{28, 41, 55, 69, 83, 96, 110, 124, 138, 152\}$. These values span a range of distinct chaotic regimes.

**Thomas (Thomas, 1999).** The Thomas attractor is a 3D strange attractor that produces cylindrically symmetric time series in state space. For this system, $M = 3$, and the dynamics are given by

$$\frac{d\mathbf{y}}{dt} = \begin{bmatrix} \sin(y_2) - by_1 \\ \sin(y_3) - by_2 \\ \sin(y_1) - by_3 \end{bmatrix} \tag{8}$$

where we sweep over $b \in \{0.025, 0.05, 0.075, 0.1, 0.125, 0.15, 0.175, 0.2, 0.225, 0.25\}$, corresponding to 10 equally spaced values from 0.025 to 0.25 in increments of 0.025. This range was chosen based on prior work (Sorin and Tulchinsky, 2024), which demonstrates significant changes in system behavior between values of 0 and 0.33.

**Hindmarsh-Rose (Hindmarsh and Rose, 1984).** This is a 3D dynamical systems model of neuronal activity that exhibits bursting behavior. For this system, $M = 3$, and the dynamics are given by,

$$\frac{d\mathbf{y}}{dt} = \begin{bmatrix} y_2 - ay_1^3 + by_1^2 - y_3 + I \\ c - dy_1^2 - y_2 \\ r[s(y_1 - x_R) - y_3] \end{bmatrix}, \tag{9}$$

and we sweep the external current parameter $I = \{1, 1.33, 1.66, 2, 2.33, 2.66, 3, 3.33, 3.66, 4\}$, corresponding to 9 equally spaced values from 1 to 4 in increments of 0.33. This range of parameters is chosen based on prior

work (Chen et al., 2023a; Dhamala et al., 2004; Goufo et al., 2020), which shows that the system exhibits different spike–burst behaviors within this region.

Given these generated time-series trials, we construct a dataset for each system, parameter setting, and noise level by combining data from five randomly selected parameter values from each system's grid. By randomly selecting these parameter values, we ensure a range of task difficulty where more challenging datasets involve classifying parameter values that are close together, whereas easier datasets involve classifying parameter values that are farther apart. Importantly, we split each trial into 70:15:15 train, validation, and test splits, and then segment each trial into non-overlapping windows of size $W = 100$. For each system and noise level, we measure the classification accuracy averaged over ten random seeds, accounting for both dataset sampling (classification difficulty) and model initialization. The results reported in Table 1 are the average result for all three systems.

## C. Baseline Description

For contrastive learning and SVAE baselines, we fix the encoder across different training objectives to control for the effects of encoder design and isolate the impact of the pre-training objective. Specifically, we adopt the time series encoder from TS2Vec (Yue et al., 2022), which consists of a 10-layer dilated convolutional network with an embedding size of 320. This setup follows the experimental protocol of (Xu et al., 2024) and report the best available performance from prior work or our own experiments. To obtain a representative embedding for each time window, we apply a global max pooling layer to aggregate features across the temporal dimension. Below, we describe each baseline in more detail.

**SimCLR (Chen et al., 2020).** SimCLR is a simple augmentation-based method that we adapt for time series data to evaluate the effectiveness of a purely augmentation-driven strategy. Three standard augmentations are applied, each with a 50% probability: scaling, which multiplies the entire time series by a factor drawn from $U(0.5, 1.5)$; shifting, which offsets the time series by a random value in the range $[-\text{subsequence size}, \text{subsequence size}]$; and jittering, which adds Gaussian noise with a standard deviation equal to 0.2 times the standard deviation of the dataset.

**TS2Vec (Yue et al., 2022).** TS2Vec is a competitive time-series contrastive learning method for time series that learns time-stamp-level representations through augmentations. It employs hierarchical contrast, combining instance-level and temporal contrast across multiple resolutions to capture scale-invariant representations within augmented context views. In our experiments,

we adopt the dilated convolutional encoder from this method and use the official implementation available at `https://github.com/yuezhihan/ts2vec`.

**REBAR (Xu et al., 2024).** REBAR is a recent time-series contrastive learning method that defines positive pairs using a learned similarity measure. This is accomplished through a cross-attention mechanism that identifies class-specific motifs in one subsequence that can be used to reconstruct another. Subsequences that have the lowest reconstruction error are selected as positive pairs for contrastive learning. We use the implementation provided in the official repository: `https://github.com/maxxu05/rebar`.

**PatchTST (Nie et al., 2022).** PatchTST is a transformer model developed for forecasting that uses a patching mechanism, in which consecutive blocks of time points are processed together, and incorporates channel independence, processing each channel separately. It achieves strong performance in forecasting tasks and has been used as a backbone for extracting information about underlying physiological states from biosignals (Geenjaar and Lu, 2025). While the original paper focuses primarily on supervised learning, the model can also be trained in a self-supervised fashion using a masked autoencoding (MAE) objective. In our experiments, we use the Hugging Face implementation from the official repository `https://github.com/yuqinie98/PatchTST` and train PatchTST in SSL mode, using the CLS token as the summary representation for downstream evaluations.

**TimeMAE (Cheng et al., 2026).** TimeMAE explores the idea of block masking in a MAE framework, adapting it to time-series data. During training, random segments of the input time series are masked, and the unmasked portions are passed through an encoder to produce latent representations. The model also introduces a decoupled autoencoder, where masked and unmasked regions are encoded separately, allowing it to extract transferable information between these regions. A lightweight decoder then reconstructs the masked segments, and the model is trained to minimize the reconstruction error. We use the implementation from the official repository: `https://github.com/Mingyue-Cheng/TimeMAE`.

**LFADS (Sussillo et al., 2016; Sedler and Pandarinath, 2023).** Latent Factor Analysis via Dynamical Systems (LFADS) is a deep generative model designed to uncover low-dimensional latent dynamics underlying neural population activity (Pandarinath et al., 2018). During inference, a bidirectional GRU produces an initial condition that serves as a summary representation of a time-series window. This initial condition is then evolved forward by a GRU with a global dynamics function to generate a latent time series, which is linearly projected back into

data space to reconstruct the original time series. Although LFADS was originally developed for neural spiking data, its Poisson likelihood can be replaced with a Gaussian likelihood to adapt the framework for continuous-valued time series. In our experiments, we modify the official codebase (`https://github.com/arsedler9/lfads-torch`) by replacing the BiGRU encoder with the same dilated convolution architecture used in previous baselines, and we find that this modification improves performance on downstream tasks.

**DSVAE (Yingzhen and Mandt, 2018).** Disentangled Sequential Variational Autoencoder (DSVAE) (Yingzhen and Mandt, 2018) is a generative model originally developed for sequential data (video and audio) and has not previously been applied to physiological signals. We include this baseline in our work since it's generative process resembles PULSE and our results show it is a competitive baseline when applied onto physiological time-series. DSVAE uses a BiLSTM and MLP to infer static and dynamic latent variables, which are then used as initial conditions and inputs to an LSTM for generation. Importantly, DSVAE does not remove irrelevant information, as both system and sample-specific latents are observed and reconstructed jointly. In contrast, PULSE explicitly discards information about noise through the its objective that leverages pseudo-pairs. We use the official implementation[3], replacing the encoder with dilated convolutions, which improves downstream performance.

**PULSE.** This is our cross-reconstruction based method for physiological time-series SSL proposed in this paper.

## D. Real Dataset Description

For the linear probe experiments, we partition the data into training, validation, and test sets with a 70/15/15 inter-subject splits. Note that the total time of the labeled subsequences may not match the full length of the original recordings, as some portions of the data can be unlabeled. Extracted subsequences are non-overlapping.

**HAR (Reyes-Ortiz et al., 2015).** The HAR dataset consists of time series data recorded from 30 volunteers of ages 19-48 years with a smartphone (Samsung Galaxy S II) attached at the subjects waist. There are 59 samples of 5-minute long time series that are collected at 50 hz. In our experiment, we use as input 6-channels ( 3-axis linear accelerometer and 3-axis angular velocity). 6-channels recordings Raw accelerometer and gyroscopic sensor data. Subsequences are 2.56 seconds long (128 time steps) which matches the labels from the original work. collected from smartphones. 6-class classification task with 4,600 sub-

---

[3]`https://github.com/yatindandi/Disentangled-Sequential-Autoencoder`

sequences with the following activity class label names and proportions: walking (17.7 %), walking upstairs (7.6 %), walking downstairs (9.1 %), sitting (18.2 %), standing (20.1 %), and laying (20.1%)

**ECG (Moody, 1983).** We use data from the MIT-BIH Atrial Fibrillation dataset (Moody, 1983). Since no subsequence length was defined in the original work, we adopt 10-second segments, following both prior analyses of this dataset (Tonekaboni et al., 2021; Xu et al., 2024) and the convention used in ECG classification studies more broadly (Wagner et al., 2020). In total, 76,590 distinct subsequences are extracted from 23 recordings, each lasting approximately 9.25 hours and sampled at 250 Hz with two channels. To further improve computational efficiency, each subsequence is downsampled by a factor of five and produces subsequences with 500 time-steps. Of these, 76,567 subsequences are labeled, with 41.7% corresponding to atrial fibrillation and 58.3% to normal rhythm.

**PPG (Schmidt et al., 2018).** This dataset is constructed from the WESAD dataset. There are 15 recordings each of approximately 87 minutes in duration, corresponding to 334,080 samples collected at 64 Hz from a single channel. From these recordings we extract 1,305 distinct 1-minute subsequences, consistent with the segmentation strategy used in the original work. We improve computational efficiency by downsampling each subsequence by a factor of four, so that each subsequence has 960 time steps. Of these, 666 subsequences are annotated with class labels: baseline (42.7%), stress (24.0%), amusement (12.4%), and meditation (20.9%). All signals are denoised following the procedure described in (Heo et al., 2021).

**EEG (Kemp et al., 2000).** Sleep-EDF (Kemp et al., 2000) contains 39 whole-night electroencephalography (EEG) recordings collected using sleep cassettes from 20 healthy subjects. Following the preprocessing protocol of (Chambon et al., 2018), we use two EEG leads (Fpz-Cz and Pz-Oz) to evaluate pretraining, sampled at 100 Hz and segmented into non-overlapping 30-second intervals (3,000 time steps). This yields a total of 35,424 samples with the following class distribution: Wake (22.9%), N1 (8.89%), N2 (51.30%), N3 (9.92%), and REM (7.00%). The data preprocessing is accomplished with the MNE package (Gramfort et al., 2013). For transfer learning, we follow prior experimental conditions and consider a pretrained model that uses a single EEG lead (Fpz-Cz), segmented into windows of length 200.

The following data are publicly available through the links provided in the repository at `https://github.com/mims-harvard/TFC-pretraining`.

**Epilepsy. (Andrzejak et al., 2001)** The Epilepsy dataset contains 500 single-channel EEG recordings, each lasting 23.6 seconds. To minimize subject-specific bias, the recordings are divided into 11,500 one-second segments and randomly shuffled, with signals sampled at 178 Hz. The dataset has five labels that capture different conditions or recording locations: eyes open, eyes closed, EEG from healthy regions, EEG from tumor regions, and seizure activity. For our experiments, we reduce the task to binary classification by grouping the first four categories as the negative class and using seizure episodes as the positive class. Fine-tuning is performed on a small labeled subset of 60 samples (30 per class), with 20 additional samples (10 per class) used for validation. The model that achieves the best validation performance is then evaluated on the remaining 11,420 test samples.

**Gesture. (Liu et al., 2009)** The Gesture dataset captures eight distinct hand movements recorded via accelerometers, with each gesture defined by the path of motion. The gestures include swiping left, right, up, or down; waving in clockwise or counterclockwise circles; tracing a square; and drawing a right arrow. Each gesture is assigned a unique classification label. While the original study reports 4,480 recordings, only 440 samples were available from the UCR Database, yielding a balanced dataset with 55 samples per class. The sampling frequency is not specified in the original paper but is assumed to be 100 Hz. Despite its modest size, the dataset provides enough samples for fine-tuning experiments.

## E. Cross Validation Experiments

Here, we further validate our findings with additional cross validation experiments that assess generalization across within-dataset variation. Specifically, we perform 5-fold cross validation, with each fold trained using a different random initialization seed. Note that we also include SimMTM (Dong et al., 2023), a recent competitive contrastive MAE hybrid model, as a baseline. Table 6 shows that PULSE achieves the best performance across all datasets, with results consistent with the single-split scores reported in Table 2. Notably, PULSE becomes the top performer on HAR when averaged over cross validation splits, even though it was not in the single-split setting. These results indicate that PULSE is robust to within-dataset variability and performs consistently across datasets with diverse signal characteristics.

The semi-supervised learning results show a similar trend. In Table 7, PULSE consistently outperforms all baseline methods, with the single exception of PPG at 1% labels, where it slightly underperforms TimeMAE. However, this difference is not significant since the accuracies are well within one standard deviation of each other. Importantly, these cross-validated results closely match the single split results in Table 3. This consistency across evaluation

*Table 6.* Linear probe classification results using 5-fold cross validation splits. Each split uses a different random seed for model initialization, and we report the standard deviation in parentheses. PULSE achieves the best performance across all datasets, and the results closely match those obtained with a single split.

| | Metric | SimCLR | TS2Vec | REBAR | PatchTST | TimeMAE | LFADS | DSVAE | SimMTM | PULSE |
|---|---|---|---|---|---|---|---|---|---|---|
| **HAR** | Accuracy ↑ | 89.47 (3.47) | 92.56 (2.96) | 94.13 (2.06) | 80.48 (1.68) | 94.16 (1.37) | 93.27 (2.44) | 92.00 (3.34) | 95.19 (1.81) | **95.30 (1.72)** |
| | AUROC ↑ | 98.31 (1.23) | 99.34 (0.36) | 99.35 (0.34) | 97.30 (0.35) | 99.59 (0.11) | 99.42 (0.33) | 99.27 (0.37) | **99.73 (0.14)** | 99.73 (0.15) |
| | AUPRC ↑ | 93.69 (3.99) | 97.51 (1.31) | 97.56 (1.09) | 89.21 (1.44) | 98.37 (0.54) | 97.83 (1.19) | 97.06 (1.45) | 98.24 (0.66) | **98.99 (0.58)** |
| **PPG** | Accuracy ↑ | 48.03 (4.16) | 57.58 (2.68) | 60.67 (3.08) | 57.71 (3.12) | 61.92 (3.64) | 55.34 (3.31) | 60.96 (2.43) | 53.93 (4.42) | **63.87 (2.50)** |
| | AUROC ↑ | 65.22 (3.06) | 76.39 (2.15) | 78.04 (2.73) | 70.93 (1.44) | 77.61 (0.86) | 73.48 (1.79) | 77.91 (3.17) | 75.74 (2.70) | **79.77 (2.77)** |
| | AUPRC ↑ | 41.89 (2.09) | 53.02 (3.91) | 57.94 (3.77) | 53.06 (1.94) | 58.27 (1.15) | 50.52 (3.40) | 57.07 (3.56) | 51.79 (3.75) | **58.43 (2.69)** |
| **ECG** | Accuracy ↑ | 73.14 (7.13) | 77.76 (4.96) | 80.00 (2.53) | 68.02 (9.18) | 67.96 (11.65) | 64.60 (3.89) | 77.93 (3.15) | 82.99 (1.72) | **88.01 (1.12)** |
| | AUROC ↑ | 75.49 (8.30) | 85.96 (7.36) | 87.19 (2.19) | 72.92 (10.98) | 87.84 (5.40) | 73.60 (8.27) | 83.30 (7.70) | 94.35 (2.31) | **96.95 (0.73)** |
| | AUPRC ↑ | 71.45 (7.31) | 80.03 (10.30) | 82.50 (0.70) | 68.99 (6.43) | 84.16 (7.81) | 67.66 (7.03) | 79.13 (9.09) | 90.49 (1.22) | **95.32 (0.29)** |
| **EEG** | Accuracy ↑ | 69.08 (2.99) | 82.15 (1.81) | 82.27 (1.74) | 80.83 (0.36) | 80.07 (0.59) | 82.37 (1.32) | 82.74 (1.75) | 80.53 (1.82) | **84.25 (1.63)** |
| | AUROC ↑ | 88.01 (1.65) | 95.01 (0.80) | 95.05 (0.73) | 94.99 (0.15) | 94.30 (0.52) | 95.00 (0.62) | 95.17 (0.60) | 94.65 (0.74) | **96.33 (0.68)** |
| | AUPRC ↑ | 57.59 (2.10) | 75.06 (1.81) | 74.73 (1.52) | 71.62 (0.12) | 74.52 (0.83) | 74.36 (1.01) | 75.00 (1.05) | 75.17 (0.94) | **77.62 (1.43)** |

*Table 7.* Semi-supervised classification accuracy for 1% and 5% of labels. Results are averaged over 5-fold cross-validation splits each with a random model initalization. Higher score is better. PULSE achieves the best performance in all settings, except for PPG at 1%, where it is within the margin of error of the top score.

| | Dataset | Supervised | SimCLR | TS2Vec | REBAR | PatchTST | TimeMAE | LFADS | DSVAE | SimMTM | PULSE |
|---|---|---|---|---|---|---|---|---|---|---|---|
| **1 %** | HAR | 81.37 (2.88) | 72.65 (1.70) | 77.91 (1.84) | 78.75 (1.74) | 33.98 (1.96) | 81.02 (1.78) | 78.54 (1.89) | 78.28 (1.62) | 79.92 (1.62) | **84.16 (2.14)** |
| | ECG | 69.20 (2.57) | 70.03 (4.50) | 74.83 (3.97) | 71.95 (4.52) | 58.31 (5.95) | 57.94 (4.79) | 66.30 (4.50) | 71.54 (6.20) | 82.02 (2.01) | **85.74 (1.30)** |
| | PPG | 43.55 (5.75) | 37.68 (4.24) | 42.83 (5.76) | 43.33 (6.05) | 42.92 (4.48) | **44.40 (4.83)** | 39.19 (3.99) | 43.24 (5.42) | 39.51 (4.51) | 43.51 (5.81) |
| | EEG | 72.32 (1.84) | 60.27 (2.04) | 75.34 (0.74) | 74.24 (1.59) | 68.32 (1.57) | 69.10 (1.43) | 71.80 (1.81) | 73.98 (1.67) | 67.41 (1.69) | **77.95 (1.26)** |
| **5 %** | HAR | 91.65 (1.35) | 82.99 (1.24) | 88.80 (1.34) | 88.64 (1.50) | 54.60 (1.89) | 91.10 (1.42) | 89.45 (1.32) | 87.38 (1.48) | 89.95 (1.33) | **92.85 (1.30)** |
| | ECG | 80.71 (3.58) | 69.64 (3.31) | 74.94 (3.46) | 71.36 (2.73) | 62.11 (6.48) | 62.37 (5.92) | 64.79 (2.57) | 73.36 (3.70) | 80.07 (2.20) | **84.95 (1.43)** |
| | PPG | 54.02 (2.24) | 42.25 (1.92) | 50.79 (2.58) | 52.74 (3.20) | 53.30 (2.43) | 45.66 (3.95) | 42.61 (2.84) | 53.75 (2.65) | 46.16 (2.86) | **54.38 (3.61)** |
| | EEG | 73.86 (2.16) | 63.38 (1.83) | 74.42 (0.50) | 73.13 (1.42) | 74.00 (1.44) | 74.76 (0.81) | 72.21 (1.75) | 72.56 (1.45) | 69.42 (1.43) | **77.87 (1.14)** |

settings indicates that PULSE's representations are stable with respect to how the data is partitioned, which further strengthens our conclusion that PULSE can learn label-efficient representations that generalize across physiological datasets with very different signal and dataset characteristics.

## F. Visualization of PULSE Representations

We visualize the embeddings produced by PULSE using t-SNE. As shown in Figure 5, PULSE learns a representation space that effectively separates the different label classes, indicating that it captures features corresponding to clinically relevant states. In the HAR dataset, PULSE separates the representations into three major super-clusters. One cluster corresponds to laying, another merges standing and sitting activities, reflecting their similar motion patterns, and the third encompasses all walking-related activities, including walking on a flat surface, walking upstairs, and walking downstairs. In ECG, there is a clear separation between A-fib from normal rhythms. Interestingly, while normal states form multiple distinct clusters, A-fib appears as a single dominant cluster, suggesting that there are many variations of normal activity, but only one typical patterns for A-fib. Clustering in PPG is less distinct, likely due to the dataset's challenging nature and the presence of motion artifacts. Nevertheless, PULSE is able to separate stress,

amusement, and meditation states, although it struggles to distinguish the baseline state. In EEG, PULSE clearly separates wake from sleep states. Furthermore, within sleep, the representation captures a continuous progression from N1 through REM states.

## G. Description of Model Training and Hyperparameters

**Pretraining.** All models are implemented in PyTorch 2.8. We pretrain using the AdamW optimizer (Loshchilov and Hutter, 2017) with a One Cycle learning rate scheduler (Smith and Topin, 2019). Regularization includes gradient clipping (fixed at 5 for all models) and weight decay, which is selected through hyperparameter tuning. Models are trained for the full number of epochs, and the checkpoint with the lowest validation loss is used for evaluation.

**Linear Probe.** We evaluate the linear probe by encoding time-series samples with a frozen model and training a Logistic Regression classifier on the standardized embeddings. We use the cuML implementation (Raschka et al., 2020), a GPU-accelerated drop-in replacement for scikit-learn. Following prior work, all hyperparameters of the linear probe are fixed, including a regularization coefficient of 1, ensuring that performance differences reflect the quality of the learned representations rather than the probe itself.

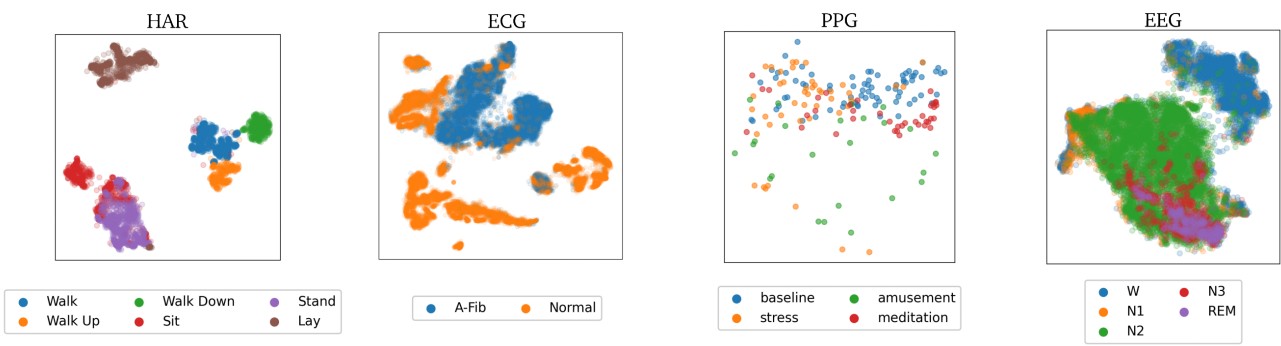

*Figure 5.* t-SNE visualizations of PULSE representations.

**Fine-tuning.** For the fine-tuning experiments, we follow the preprocessing and dataset-splitting procedure of (Zhang et al., 2022). We initialize the encoder with the best pre-training checkpoint and attach a two layer fully connected classification head with a hidden dimension of 64 and an output dimension matching the target dataset. Fine tuning is performed for 40 epochs using a batch size of 30, a learning rate of 0.0003, and a weight decay of $10^{-5}$.

**Hyperparameters.** We tune hyperparameters for each method using a random search with a budget of 30 trials over a grid of reasonable values. For trial, we uniformly sample within each of the following grids and select the best setting according to the best validation performance. Training hyperparameters are swept over epochs $[50, 100, 200]$, learning rates $[0.001, 0.0005, 0.0001]$, and weight decay $[10^{-3}, 10^{-4}, 10^{-5}]$. Model-specific hyperparameters are varied to capture differences in architecture and training objectives.

For PULSE, the model-specific hyperparameter sweep includes the initial condition encoder hyperparameters, including convolution kernel size $[3, 5, 11]$, dilation $[1, 2]$, and hidden dimensionality $[64, 128]$. For the GRU decoder, we sweep across the number of layers $[2, 3, 4]$ and hidden dimensionality $[64, 128]$. Finally, for the pseudo-pair construction, we vary the number of samples $[1, 2, 3]$. Hyperparameter grids for other baselines are included in the code repository.

## H. Potential Mechanisms for Learning Well-Structured Representations

PULSE produces a structured representation space without the explicit mechanisms used by contrastive learning (CL). We discuss why this structure may emerge and how the two approaches differ.

**How cross-reconstruction shapes the learned representation.** In principle, our pseudo-pair strategy could learn separate system parameters for every window. In practice,

this doesn't happen and we observe a well-structured latent space in which windows with similar dynamics cluster together (fig. 5). We hypothesize that the pretraining objective implicitly encourages smoothness. To cross-reconstruct a randomly sampled target, it may be more stable to optimize for a well-organized space that lets the decoder reuse shared components; scattering similar signals arbitrarily would place the parameters needed for reconstruction in a distant neighborhood, yielding high error. This would encourage the model to relate shared dynamics across windows rather than learn window-specific dynamics, consistent with the transfer across windows and to held-out subjects that we observe.

**Distinctions from contrastive learning.** Both approaches yield structured representations but rely on different mechanisms. While CL explicitly minimizes the distance between positive pairs, PULSE groups similar samples implicitly, since nearby signals let the decoder reuse shared dynamical components. While CL explicitly separates negative samples, PULSE does not require negative samples, as mapping two distinct systems to the same representation leaves the decoder unable to determine which to reconstruct, again yielding high error. Finally, PULSE is generative, preserving dynamical information through cross-reconstruction, whereas CL is discriminative and may discard these dynamics when they are unnecessary for distinguishing samples.

**Toward independent pairs.** Although the pseudo-pair strategy approximates properties of independent pairs, our synthetic experiments suggest truly independent pairs yield further gains. Identifying or constructing such pairs from raw, unlabeled data is a promising direction for future work.

## I. Method Limitations

An important limitation of our work is the gap between the practical PULSE algorithm and the theoretically ideal cross-reconstruction setting. While Theorem 1 suggests

using fully independent samples to learn system representations invariant to sample-specific factors, PULSE relies on pseudo-pairs from the same sample. This may limit the model's ability to capture system dynamics, particularly when sample-specific variability is large. Future work could improve the identification of independent samples to better align practice with theory. Another limitation is the structure of our assumed generative model. Although it captures key system-level and sample-specific components, it may not fully reflect the complexity of real physiological signals. Future work in this direction could explore more flexible or data-driven generative models to improve the fidelity and expressiveness of learned system representations.

