# OpenReview forum: "Self-Supervised Dynamical System Representations for Physiological Time-Series"
_ICML.cc/2026/Conference — ICML 2026 regular_

### Official Review · Reviewer_4rJL · 2026-02-16

**Soundness:** 3
**Presentation:** 3
**Significance:** 3
**Originality:** 3
**Overall Recommendation:** 5
**Confidence:** 3

**Summary:**

The author proposes a self-supervised learning method to pretrain the backbone model on physiological signals leveraging the underlying dynamical information in the observed time series. Theoretical arguments and visualization are provided to support the proposed idea. The proposed model, PULSE, is evaluated on multiple datasets covering different sensing modalities, and demonstrates leading performance compared to the chosen baselines.

**Compliance With Llm Reviewing Policy:**

Affirmed.

**Final Justification:**

We thank the author for the comprehensive response with solid additional empirical evidence to support the main claimed contributions. All of my concerns are addressed, and the confusions are cleared with straightforward clarification. I will updated my rating accordingly.

**Key Questions For Authors:**

The pretraining dataset information is missing. Is PULSE pretrained on a synthetic generated dataset or disjoint real-world datasets that are left out with those used for evaluation?

**Limitations:**

Yes.

**Strengths And Weaknesses:**

# Strengths
- The idea of learning the dynamical representation lies under the physiological signals is a very interesting direction.
- The argument behind the methodology is supported by well-rounded theoretical arguments.
- Visualizing the physiological signal as a dynamical system is a very inspiring idea. To the best of my knowledge, this is the first of this kind of work that approaches physiological signal modeling from a dynamical system point of view.
- The downstream evaluation shows that the proposed model outperforms the chosen baselines significantly.

# Weaknesses
- The choice of baseline is not very physiological time series focused, where most of the foundation models in literature for physiological signals are not included, for example [1-4]. Having a comparison with these kind of baselines could better contextualize the meaning of the performance of the proposed method.
- Though a solid theoretical argument is provided, the high-level idea behind the proposed method seems similar to contrastive learning, where the contrastive loss aims to tighten the representations of samples that share similar labeled features or hidden features, and push away samples that do not share those features. More discussion or comparison around this aspect could help reader to better understand the core novelty lies in the proposed idea.
- Not actually a weakness, but more like a suggestion: visualizing the physiological time series in a vector field with trajectory being highlighted is visually impactful. A more polished description of how these plots are generated and the meaning of the vector directions are expected for clarity. And presenting more of this kind of visualization (e.g. in a grid field with multiple signal samples and categorized by different type of signal such as ECG, EEG, abnormal signal, normal signal, etc.).

[1] NormWear, for encoding arbitrary physiological signal data: Luo, Yunfei, et al. "Toward foundation model for multivariate wearable sensing of physiological signals." (2024).

[2] Pillai, Arvind, et al. "Papagei: Open foundation models for optical physiological signals." (2024).

[3] ECG-FM, for encoding ECG data: McKeen, Kaden, et al. "Ecg-fm: An open electrocardiogram foundation model." (2025):.

[4] CBraMod, for encoding EEG data: Wang, Jiquan, et al. "Cbramod: A criss-cross brain foundation model for eeg decoding." (2024).

---

> ### Author Rebuttal · Authors · 2026-03-30
>
> Thank you for your comments. We appreciate that you find our work **interesting, theoretically well-supported**, and **inspiring**.
>
> **W1:** **PULSE is NOT a Foundation Model (FM), and direct comparisons with them are confounded by many factors.** Our evaluations are designed to isolate the effect of the pretraining objectives by controlling for the data, model size, and compute. However, **we agree that comparisons with FMs could provide valuable context for the community.**
>
> Below, **we present comparisons with 5 new FM baselines**, including 2 general-purpose models (MOMENT-L [1] and NormWear [2]) and 3 modality-specific models (ECG-FM [3], PaPaGei [4], and CBraMod [5]). For general-purpose FMs, we report linear probe results on frozen FM representations across all datasets. For modality-specific FMs, we report only on the relevant datasets. For fairness, we interpolate the sampling rate to match the original training settings of each FM.
>
> **PULSE demonstrates strong performance, either matching (HAR/ECG) or outperforming (PPG/EEG) all recent FMs** using only a fraction of the data and compute resources. Specifically, our encoder is only ~650k parameters (which is 0.1% of parameters in MOMENT), demonstrating that PULSE is highly parameter efficient without loss in performance. This validates that PULSE is a highly effective pretraining strategy.
>
> **Table. Comparisons with FMs for Linear Probe.**
>
> | Dataset | Metric | MOMENT | NormWear | ECG-FM | PaPaGei | CBraMod | **PULSE (Ours)** |
> | :--- | :--- | :---: | :---: | :---: | :---: | :---: | :---: |
> | **HAR** | ACC / AUROC | 67.61 / 91.69| 93.01 / 99.35 | — | — | — | **93.27** / **99.42** |
> | **ECG**| ACC / AUROC | 80.27 / 94.42  | 75.13 / 86.01 | **88.20** / 94.37 | — | — | 87.41 / **94.93** |
> | **PPG**| ACC / AUROC | 58.43 / 73.80 | 53.93 / 72.90 | — | 49.43 / 73.31 | — | **64.27** / **80.29** |
> | **EEG**| ACC / AUROC | 77.78 / 93.79 | 73.95 / 93.35 | — | — | 49.22 / 81.03 | **85.56** / **96.17** |
>
>
> [1] MOMENT (Goswami et al. 2024)
>
> [2] NormWear (Luo et al. 2026)
>
> [3] ECG-FM (McKeen et al. 2025)
>
> [4] PaPaGei (Pillai et al. 2025)
>
> [5] CBraMod (Wang et al. 2025)
>
> &nbsp;
>
> **W2:** We appreciate the opportunity to clarify our novelty. **PULSE and contrastive learning (CL) rely on fundamentally different learning mechanisms**, which we discuss below:
>
> **Grouping similar samples.** Unlike CL, which explicitly minimizes the distance between positive pairs, PULSE implicitly groups similar samples together to improve cross reconstruction performance. If the representation space brings similar signals together, then the decoder can reuse shared dynamical components to effectively reconstruct randomly sampled pairs. In contrast, if similar signals were mapped to completely different regions, the reconstruction error would be high, because the optimal parameters required for successful reconstruction of a randomly sampled target could exist in a completely different neighborhood than what was originally estimated.
>
> **Separating dissimilar samples.** Unlike CL, which explicitly separates negative samples, PULSE learns well-structured representation spaces without the use of negative samples. Instead, separation occurs implicitly because if two distinct systems are mapped to the same representation, the decoder cannot distinguish which system to reconstruct, therefore resulting in high reconstruction error.
>
> **Generative vs Discriminative.** PULSE is a generative task designed to preserve dynamical information through a structured cross-reconstruction task. In contrast, CL is a discriminative task that may arbitrarily discard these dynamics if they are not strictly necessary for distinguishing between samples.
>
> Thank you for raising this point. **We agree that this discussion is valuable and we will update our manuscript to include these insights.**
>
>
> &nbsp;
>
>
> **W3:** This is a great suggestion! We agree these visualizations would be impactful. Unfortunately, our learned systems are time-varying and high-dimensional (64D), making it unclear how to visualize in a 2D plot. Since visualizing high-d dynamics is an active area of research [1,2] often restricted to simple scenarios, this would require nontrivial method development. Our goal is representation learning, not mechanistic interpretability, and these plots are only straightforward for 2D systems.
>
> [1] Reverse engineering RNNs with Jacobian SLDS. (Smith et al. 2021)
>
> [2] Mechanistic Interpretability of RNNs emulating HMMs (Torre et al. 2025)
>
>
> &nbsp;
>
>
> **Q1:** PULSE is pretrained and evaluated on the same dataset using a **strictly disjoint, subject-wise split**, without external or synthetic data. This follows the health time-series SSL literature, where all subjects used for testing were entirely excluded from the pretraining phase. Thus, the reported metrics reflect the ability to effectively transfer to **unseen individuals** within the same recording context. **We will make this clear on revision.**

---

> > ### Author Rebuttal · Reviewer_4rJL · 2026-04-02
> >
> > We thank the author for the comprehensive response with solid additional empirical evidence to support the main claimed contributions. All of my concerns are addressed, and the confusions are cleared with straightforward clarification. I will updated my rating accordingly.

---

### Official Review · Reviewer_Lrrd · 2026-02-18

**Soundness:** 3
**Presentation:** 3
**Significance:** 3
**Originality:** 3
**Overall Recommendation:** 5
**Confidence:** 2

**Summary:**

This paper proposes learning dynamical systems to represent time series in a semi-supervised setting, while disentangling sample-specific noise from transferable information. The key idea is to share latent dynamical parameters across subgroups of time series, capturing common underlying dynamics, while maintaining independent latent initializations for each sample and using a shared observation function across the entire dataset. The learning mechanism is based on a cross-reconstruction between pseudo-pair that has been theoriticaly motivated.

Extensive experiments on both synthetic and real-world datasets—including ECG, HAR, EEG, and PPG—demonstrate the effectiveness of the approach for classification tasks with both small and large training sets, as well as for transfer learning tasks on EEG and HAR data.

**Compliance With Llm Reviewing Policy:**

Affirmed.

**Key Questions For Authors:**

Can you address my doubt about significance ?

**Limitations:**

The authors have stated limitations and societal impact.

**Strengths And Weaknesses:**

Soundness:
The paper is technically solid, both from a theoretical and an experimental standpoint. The cross-reconstruction loss is well motivated and carefully compared to alternative losses. The experiments are conducted across multiple random seeds, different datasets, and various tasks, with fair and appropriate baselines.

Presentation:
Overall, the paper is clearly written. I only noted two somewhat clunky points: the evaluation protocol using a linear probe is not clearly specified for the synthetic data, and the sentence “we our outline assumptions” (p.5) appears to contain a typo.

Regarding the related work, it would have been interesting to broaden the perspective on group-structured modeling by discussing connections with mixed-effects models or federated learning. For instance, the gait analysis paper "Personalized Convolutional Dictionary Learning of Physiological Time Series" could provide a relevant reference point.

Significance:
The paper positions itself as a compelling combination of structurally constrained models and contrastive learning, aiming to preserve the right amount of sample-specific information while learning transferable dynamics. Both the theoretical developments and the experimental results are convincing.

However, I still have some doubts about the notion of “system information” in Section 3.1. When reading the paper, I initially expected a mixture-based approach, where different windows or recordings would be grouped under the same system. Instead, if I understand correctly, there seem to be as many systems as windows, since all pseudo-pairs are derived from the same window.

To summarize my understanding: the measurement function is shared across all windows; the latent initialization is sample-specific (where a sample corresponds to a cropped window); and the system parameters are window-specific. Am I misunderstanding something?

If this interpretation is correct, it raises the question of whether the observed performance truly supports the claim of transferability, since the notion of transfer appears rather narrow in this setting.

Originality:
The paper is original in the way it combines well-established mechanisms into a novel methodological framework.

---

> ### Author Rebuttal · Authors · 2026-03-30
>
> Thank you for your highly supportive feedback and careful reading. We are encouraged that you find our work **novel, clearly written**, and **technically solid both theoretically and experimentally**. Your insights have helped us better articulate our framework, which further improves our work. Please see our discussion on significance below:
>
> &nbsp;
>
> **Concern 1 (Synthetic Experiment Details and Typos):** Thank you for your attention to detail! We will correct these typos and update the manuscript to include more details for the synthetic evaluations.
>
> &nbsp;
>
> **Concern 2 (Connection to group-structured modeling):**  Thank you for this insight. We agree that the hierarchical formulation in our SSL framework shares a clear motivation with mixed-effects models, where 'fixed effects' represent shared system dynamics and 'random effects' capture the sample-specific variability within an observed time-series window. **We will update our Related Works to better contextualize PULSE within the established field of mixed-effects modeling**, by discussing connections to [1-3] and any other relevant literature.
>
> [1] Personalized Convolutional Dictionary Learning of Physiological Time Series (Roques et al. 2025)
>
> [2] Mixed Effects Neural Networks (MeNets) with Applications to Gaze Estimation. (Xiong et al. 2019)
>
> [3] Mixed-Effect Time-Varying Network Model and Application in Brain Connectivity Analysis. (Zhang et al. 2019)
>
> &nbsp;
>
> **Concern 3 (Significance):** Your understanding is correct and it is theoretically possible to learn completely separate system parameters for every single window when using our pseudo-pair strategy. **However, this doesn’t happen in practice.** In our experiments, we observe the emergence of a well-structured latent space (Fig. 5), where windows with similar dynamics naturally cluster together.
>
> We hypothesize that this is because our pretraining objective implicitly encourages smoothness in $\theta$ space. In order to successfully cross-reconstruct a randomly sampled target, **it may be more stable to optimize for a well-organized $\theta$ space that allows the decoder to leverage and reuse shared components.** Conversely, if the encoder were to scatter similar signals arbitrarily in representation space, the optimal $\theta$ required to successfully reconstruct a randomly sampled target could exist in a completely different neighborhood, thereby resulting in high reconstruction error.
>
> This implicit mechanism encourages the model to relate shared dynamics across windows instead of learning window-specific dynamics, which is what enables effective transfer across different windows and to held-out subjects in our experiments. While our pseudo-pair strategy is effective at approximating certain properties of independent pairs, it may be possible to achieve further performance gains by using truly independent pairs as demonstrated by our synthetic experiments. This suggests an exciting direction for future work that develops methods that more effectively identify or construct independent pairs from raw, unlabeled datasets.
>
> Thank you for raising this point. **We agree that this discussion is valuable and we will update our manuscript to include these insights.**

---

> > ### Author Rebuttal · Reviewer_Lrrd · 2026-04-01
> >
> > Thank you for addressing my concerns.

---

### Official Review · Reviewer_o84D · 2026-03-11

**Soundness:** 3
**Presentation:** 3
**Significance:** 3
**Originality:** 2
**Overall Recommendation:** 4
**Confidence:** 4

**Summary:**

This work proposes a novel pre-training framework for capturing dynamical relationships while selectively removing irrelevant noise by integrating a dynamic system into a cross-reconstruction framework. Relevant theories are explored and comprehensive experiments are conducted on multiple datasets for validation.

**Compliance With Llm Reviewing Policy:**

Affirmed.

**Final Justification:**

The rebuttal has addressed my main concerns.

**Key Questions For Authors:**

1. PULSE discards sample-specific information that is not transferable. How does the model ensure that individual-specific characteristics can still be captured during downstream fine-tuning? This is particularly critical in medical applications, where patient-specific features can have significant clinical implications.
2. What is the motivation for slicing time-series data to construct new datasets? For modalities such as ECG and EEG, most pretraining methods operate on entire recordings. Why is this approach necessary here? Is it because the proposed method cannot effectively handle long sequences? If so, what are the current limitations or bottlenecks of the model when processing extended time-series data?
3. How is  𝑆 (the total number of systems) determined? How are samples assigned to their corresponding set? The manuscript does not specify the clustering or assignment strategy, which represents a critical gap at the implementation level.
4. Regarding the selection of sample pairs: how can “similar sample pairs” be constructed efficiently? Are pairs generated by randomly sampling samples from the same system, or is a more refined clustering or matching strategy required to ensure the quality of the pairs?
5. Should process noise be encoded separately? If the noise is systematic, such as sensor bias, does it also constitute a form of “transferable information”?

**Limitations:**

Yes

**Strengths And Weaknesses:**

Strengths:
1. A novel framework designed to overcome the limitations of MAEs and SVAEs.  The proposed PULSE leverages dynamic systems to model the underlying physical and biochemical constraints of physiological signals.  Moreover, by employing a pseudo-pair strategy combined with a cross-reconstruction task, it effectively addresses the tendency of SVAEs to incorporate noise into the latent representation.
2. The article proves the necessary conditions for system information recovery by providing the analysis of Theorem 1 and related synthetic experiments, thereby supporting the design motivation of this paper.
3. Extensive experiments were conducted on different physiological signal datasets and good performance was achieved.
3. Even with only 1% or 5% of labels for training, PULSE still maintains an extremely high classification accuracy rate, which is crucial for clinical applications where labeling costs are high.

Weaknesses:
1. Theorem 1 suggests using completely independent samples, but the actual PULSE algorithm relies on pseudo-pairs from the same record, which may limit the model's ability to capture system dynamics when the intra-sample variation is extremely large.
2. The effectiveness of cross-reconstruction critically depends on accurately identifying samples that share the same system parameters. As shown in the ablation study, if the pseudo-pair strategy fails, performance drops substantially (by approximately 9.68%). This suggests that the primary advantage of PULSE arises not directly from its architecture, but from this specific sampling technique. Under certain acute pathological conditions—such as sudden arrhythmias or epileptic seizures—physiological system parameters can change dramatically within a very short time. In such cases, adjacent windows from the same recording may exhibit entirely different dynamic characteristics, and enforcing cross-reconstruction can introduce significant deviations. This may lead the model to underperform relative to its potential.
3. The computational overhead of complex dynamic encoders and cross-reconstruction mechanisms on extremely large-scale datasets may be higher than that of simple contrastive learning methods.
4. The baselines used for comparison are all from 2023 or earlier, lacking more recent studies from 2024 and 2025. This limits the persuasiveness of the experimental results.

---

> ### Author Rebuttal · Authors · 2026-03-30
>
> Thank you for your feedback. We appreciate that you consider our work to be **novel, theoretically supported**, and **supported by extensive experiments**. Please see our responses below.
>
> &nbsp;
>
> **W1:** Large intra-sample variation is a core challenge in physiological signals that is not unique to our method. **PULSE addresses intra-sample variability through a theory and model specifically designed to account for nonstationarity within recording windows**. Theorem 1 is broad enough to apply to certain time-varying dynamics, and provides the basis for modeling time-varying variables (section 3.2) that facilitate learning despite significant intra-sample shifts. Our PPG experiments, containing large motion artifacts, demonstrate that PULSE outperforms existing methods despite large intra-sample variation.
>
> &nbsp;
>
> **W2:** We apologize for the confusion, but this is an incorrect interpretation of our method. **PULSE pseudo-pairs are NOT obtained by sampling “adjacent windows”.** Rather, they are constructed from a single window as described in section 3.2. As a result, adjacent windows are free to contain completely different semantics that may arise due to rapid transitions from acute conditions. **We will clarify our method description on revision.**
>
> Furthermore, the suggestion that our proposed architecture does not contribute to the performance gains is not supported by the presented data.  Specifically, our ablation study shows that removing key architecture, such as time-varying parameters, results in a significant performance drop (7.62%). Thus, **both the pseudo-pair strategy AND the architecture are important**. The pseudo-pairs provide the necessary training signal, while the architecture and pretraining task effectively learns from that signal.
>
> &nbsp;
>
> **W3:** Please see our **new runtime comparisons in our response to Reviewer PMDr (W2)**. We show that PULSE runtime is reasonable and that any additional overhead introduced is very lightweight.
>
> &nbsp;
>
> **W4:** Please see our **new experiment in our response to Reviewer 4rJL (W1).** We compare with several foundation models (FMs) from 2024-2026. Overall, PULSE either matches or outperforms these FMs with a fraction of the compute.
>
> &nbsp;
>
> **Q1:** Sample-specific noise and individual-specific physiology are fundamentally different. **PULSE does not explicitly discard individual-specific characteristics,** such as a unique signal morphology due to individual physiology, because these features persist across multiple windows per subject. Instead, PULSE targets the removal of sample-specific noise that is unique to a single window. This noise does not transfer between windows, and does not reflect the underlying physiology. By removing non-physiological features, PULSE can prioritize individual-specific characteristics needed for clinical applications.
>
> &nbsp;
>
> **Q2: Slicing recordings into windows is a deliberate preprocessing step, and is NOT due to compute limitations.** Slicing aligns the input data with the specific time-scale of relevant clinical states. **Window sizes are determined by domain experts** to ensure that each segment captures sufficient context for a particular task. For example, the American Academy of Sleep Medicine (AASM) defines one sleep stage label for every 30s epoch because EEG activity is considered consistent within that period for sleep [1]. In contrast, summarizing an entire multi-hour recording into a single label for sleep staging is clinically inappropriate since it would collapse distinct sleep states together. Contrary to the reviewer's suggestion, windowing is a standard practice and **most pretrained models rely on segmented data** (i.e. ECG-FM (5s windows) [2], SleepFM (5s) [3], and CBraMod (30s) [4]).
>
> [1] The AASM manual. (Iber 2007)
>
> [2] ECG-FM. (McKeen et al. 2025)
>
> [3] CBraMod (Wang et al. 2025)
>
> [4] SleepFM. (Thapa et al. 2026)
>
> &nbsp;
>
> **Q3: S is not a hyperparameter that we explicitly determine, nor are samples manually assigned to a corresponding set.** Instead, S is a structural parameter used only to define our graphical model which is necessary for formulating Theorem 1. **It is not a variable required during the SSL pretraining.** Our pretraining implicitly clusters similar dynamics due to the cross-reconstruction task (see Reviewer 4rJL, W2).
>
> &nbsp;
>
> **Q4:** We apologize for the confusion. We do not "select" or "match" sample pairs. Instead, PULSE pseudo-pairs are constructed by taking a random crop of the original window (section 3.2). **This is a highly efficient operation** that does not require any clustering strategies. **We will make this clear  on revision.**
>
> &nbsp;
>
> **Q5:** We recommend removing any noise artifacts via preprocessing that do not reflect the underlying physiological state if possible. Even if noise is systematic, it is a confounding factor rather than a transferable clinical signal, as these biases are independent of the physiology of interest.

---

> > ### Author Rebuttal · Reviewer_o84D · 2026-04-02
> >
> > Thanks for addressing my concerns. I will raise my score to 4.

---

### Official Review · Reviewer_PMDr · 2026-03-12

**Soundness:** 2
**Presentation:** 2
**Significance:** 2
**Originality:** 2
**Overall Recommendation:** 3
**Confidence:** 3

**Summary:**

This paper proposes PULSE, a self-supervised pretraining method for physiological time series. By combining dynamical system modeling with cross-sample reconstruction, it is theoretically characterized and empirically validated on both synthetic and real-world datasets to better disentangle transferable system dynamics from sample-specific noise. As a result, PULSE significantly improves representation quality, label efficiency, and transfer learning performance.

**Compliance With Llm Reviewing Policy:**

Affirmed.

**Key Questions For Authors:**

see **Weaknesses**

**Limitations:**

yes

**Strengths And Weaknesses:**

**Strengths**

1. The authors addressed limitations of existing self-supervised learning methods for physiological time series, particularly their difficulty in separating shared, transferable dynamical patterns from individual-specific noise or nuisance factors.

2. They offered clear and plausible theoretical motivation for their PULSE algorithm.

3. The authors tested PULSE's performance and compared it with previous methods.


**Weaknesses**

1. PULSE relies on the presence of learnable latent dynamical structure, which may not hold for highly irregular physiological signals.

2. The joint design of dynamical modeling and cross-sample reconstruction introduces additional computational and optimization overhead.

3. Cross-sample reconstruction may be less effective when subjects differ substantially in demographics, acquisition settings, or clinical states.

---

> ### Author Rebuttal · Authors · 2026-03-30
>
> Thank you for your time and feedback. We are encouraged that you find our work **theoretically well-motivated** and that it **addresses existing limitations**. We address each weakness below:
>
> &nbsp;
>
> **W1:**
> We agree with the reviewer that physiological signals are challenging to work with and require special considerations. However, **we respectfully disagree with the premise that challenging time-series properties imply a lack of learnable dynamical structure**. This is because physiological time-series are the product of highly structured biological/physiological constraints, rather than due to unstructured randomness. As a result, **there is a long history of research dedicated to modeling a broad range of “irregular” health time-series with dynamical systems [1-8]**. These works demonstrate that it’s possible to model dynamical structure despite very complex time-series behaviors.
>
> We find it difficult to respond to this criticism beyond general pointers to the literature showing that this is an active research area producing results, since it implies that the whole field is irrelevant which is untrue due to the citations above. This concern is not specific to PULSE and in fact, **our work demonstrates that PULSE can successfully learn structure from challenging datasets with “irregular” properties**. In section 4, we show strong performance on stochastic chaotic systems which produce noisy and non-repeating time-series. In section 5, we show that PULSE captures the underlying physiological state in nonstationary signals like EEG and signals containing significant artifacts like PPG. **If the reviewer has any specific concern about how the PULSE framework is particularly vulnerable to this concern, we are happy to reply in more detail.**
>
> [1] Identification of nonstationary dynamics in physiological recordings. (Kohlmorgen et al 2000)
>
> [2] Nonlinear dynamic modeling of physiological systems. (Marmarelis 2004)
>
> [3] Time Series Analysis of Complex Dynamics in Physiology and Medicine (Glass and Kaplan 1993)
>
> [4] Nonlinear dynamics of cardiovascular ageing. (Shiogai et al. 2009)
>
> [5] Human movement variability, nonlinear dynamics, and pathology: is there a connection? (Stergiou et al. 2011)
>
> [6] Dynamic models of large-scale brain activity (Breakspear 2017)
>
> [7] Neural population dynamics during reaching (Churchland 2012)
>
> [8] Inferring single-trial neural population dynamics using sequential auto-encoders. (Pandarinath et al. 2018)
>
> &nbsp;
> &nbsp;
>
>
> **W2:**
> Thank you for raising this concern about compute efficiency. To address this, **we present new runtime benchmarks below**.
>
> The following table reports the training step latency (combined forward and backward pass) on a batch size of 128 on a single NVIDIA RTX A5000. Results are averaged over 100 repeats after discarding the first 20 “warm up” iterations. We report both HAR and PPG to represent performance on short (128 samples) and long (3,840 samples) time-series.
>
> **Table: Runtime (s) per Batch of  Size 128**
> | Method | HAR (s) | PPG (s) |
> | :--- | :--- | :--- |
> | SimCLR | 0.0489 | 0.0286 |
> | TS2Vec | 0.0574 | 0.2960 |
> | timeMAE | 0.1338 | 0.6702 |
> | LFADS | 0.6224 | 3.8842 |
> | PULSE | 0.0798 | 0.2461 |
>
> As shown, the **PULSE runtime is reasonable** and falls well within the range of existing baselines. PULSE is significantly faster than timeMAE and LFADS, and it achieves a runtime comparable to TS2Vec. This positions our approach as an efficient alternative to existing reconstruction-based baselines and shows that **any additional overhead we introduce is very lightweight**.
>
> &nbsp;
> &nbsp;
>
> **W3:** Our experiments (section 5), which were conducted with disjoint inter-subject dataset splits, show that **PULSE leads to more transferable representations to unseen subjects than current SOTA approaches**. Subject variability is a fundamental challenge in working with physiological time-series that is not unique to our cross-reconstruction approach.
>
> We hypothesize that **PULSE’s improvements over existing methods are due to specifc modeling advantages when addressing subject variability**. For instance, contrastive learning (CL) often inappropriately separates samples with shared physiological semantics from different subjects due to false negatives, which prevents the learned representations from transferring effectively between different subjects. In contrast, cross-reconstruction avoids this problem by eliminating the notion of negative samples entirely, allowing for the possibility to learn shared representation between different subjects as long as they generate time-series with similar dynamics.

---

> > ### Author Rebuttal · Reviewer_PMDr · 2026-04-04
> >
> > I will keep my score.

---

### Decision · Program_Chairs · 2026-04-30

**Decision:**

Accept (regular)

**Comment:**

PULSE introduces a new way to learn representations for health signals. Most models fail to separate shared patterns from random noise. PULSE fixes this. It uses a dynamical systems model to find what is transferable between samples. It employs a cross-reconstruction task with pseudo-pairs. This lets the model learn the core system dynamics while it ignores sample-specific noise. The authors back this with a theoretical proof. They also show that PULSE is small and fast. It matches or beats large foundation models with far less compute.

Reviewers Lrrd and 4rJL find the work strong. They like the clear theory and the wide range of tests. They see the dynamical system view as an inspiring path for this field. Reviewer o84D also finds the work solid. He noted some risks with the pseudo-pair strategy, but the results still convince him. Reviewer PMDr raised concerns about compute costs and signal irregularity. The authors answered these points with new data and literature. PMDr acknowledged that these concerns were fully resolved.